# Improved seeding strategies for `k-means` and `k-GMM`

**Guillaume Carrière**                                                        *guillaume.carriere@inria.fr*
*Centre Inria at Université Côte d'Azur, France*

**Frédéric Cazals**                                                              *frederic.cazals@inria.fr*
*Centre Inria at Université Côte d'Azur, France*

**Reviewed on OpenReview:** *https://openreview.net/forum?id=4Ut2YnekhN*

## Abstract

We revisit the randomized seeding techniques for `k-means` clustering and `k-GMM` (Gaussian Mixture model fitting with Expectation-Maximization), formalizing their three key ingredients: the metric used for seed sampling, the number of candidate seeds, and the metric used for seed selection. This analysis yields novel families of initialization methods exploiting a *lookahead* principle–conditioning the seed selection to an enhanced coherence with the final metric used to assess the algorithm, and a *multipass strategy* to tame down the effect of randomization.

Experiments show a significant improvement over classical contenders. In particular, for `k-means`, our methods improve on the recently designed multi-swap strategy (similar results in terms of SSE, seeding $\sim \times 6$ faster), which was the first one to outperform the greedy `k-means++` seeding.

Our experimental analysis also shed light on subtle properties of `k-means` often overlooked, including the (lack of) correlations between the SSE upon seeding and the final SSE, the variance reduction phenomena observed in iterative seeding methods, and the sensitivity of the final SSE to the pool size for greedy methods.

Practically, our most effective seeding methods are strong candidates to become one of the–if not the–standard techniques. From a theoretical perspective, our formalization of seeding opens the door to a new line of analytical approaches.

## 1 Introduction

**The `k-means` and `k-GMM` problems.** Clustering with `k-means` and designing Gaussian mixture models (GMM) with `k-GMM` algorithms play a pivotal role in unsupervised analysis. Consider a point set $X = \{x_1, \ldots, x_n\} \subset \mathbb{R}^d$, to be partitioned into a predefined number $K$ of clusters, or to be modeled as a mixture of $K$ multivariate Gaussian distributions.

A `k-means` clustering is a hard partition of the $n$ points into $K$ clusters, each consisting of the data points located in a Voronoi region of the Voronoi diagram of $K$ centers – which in general are not data points. The quality of the partition/clusters is assessed by the Sum of Squared Errors (SSE) functional, namely the sum of squared distances between a point and its nearest center. The search space of `k-means` is therefore the space of partitions of the point set. For fixed $K$ and $d$, the number of partitions induced by Voronoi/power diagrams is polynomial (Inaba et al., 1994). Alas, the corresponding algorithm is not practical, which motivated the development of so-called Lloyd iterations that iteratively update a set of initial centers (Lloyd, 1982). A related problem consists of designing a mixture of $K$ multivariate Gaussian distributions, so as to maximize the likelihood of the points. The search space is now the parameter space of these Gaussian distributions, yielding a more challenging endeavor. One pivotal technique to design such mixtures is the Expectation-Maximization algorithm (Dempster et al., 1977; Wu, 1983), an iterative process refining an initial guess. It may be observed that `k-means` performs a hard clustering, while `k-GMM` provides a *responsibility* of each

component for each point, which may be seen as a soft assignment. Interestingly, `k-means` can be derived as a limit case of EM (Bishop & Nasrabadi, 2006).

**Seeding strategies for `k-means` and `k-GMM`.** Both Lloyd iterations and EM are iterative methods heavily relying on the starting point, namely the initial centers in `k-means`, and the initial Gaussian components in `k-GMM`. In order to reduce the number of (Lloyd, EM) iterations, the init step seeks seeds as representative as possible of the final result. This is practically done in an iterative fashion, for $k = 1, \ldots, K$. To define the $k$-th seed/component, one uses point(s) not well described by the previously chosen/defined $k-1$ seeds/components, and ideally provides a good representative in a $k$-components model. A vast array of techniques have been explored, both for `k-means` (Celebi et al., 2013) and `k-GMM` (Kwedlo, 2013; Blömer & Bujna, 2016; You et al., 2023). These seeding strategies recently underwent important developments consisting of improving the initial $K$ seeds via a *re-selection* mechanisms, local searches and swaps (Lattanzi & Sohler, 2019), (Choo et al., 2020), (Fan et al., 2023), (Grunau et al., 2023).

**Contributions.** We design novel seeding methods for `k-means` and `k-GMM` yielding (i) a lesser number of (Lloyd, EM) iterations, and (ii) a lesser variability in the output. To achieve these goals, we revisit the previous seeding methods and formalize their three key ingredients: the metric used to sample candidate seeds, the number of seed candidates, and the metric used to rank candidate seeds. This analysis brings out two general design principles for seeding methods. The first is a *lookahead* principle, which consists of conditioning the seed selection to an enhanced coherence with the final metric used to assess the algorithm. The second is a *multipass strategy*, which consists of performing the seed selection in at least two passes, to tame down the effect of randomization.

Our methods bear two major differences with the recently developed reselection schemes (Lattanzi & Sohler, 2019; Choo et al., 2020; Fan et al., 2023; Grunau et al., 2023; Huang et al., 2024). The first is the *metric* used to perform selection, which is the distance to the centroids of the clusters induced by the centers–instead of the distances to the centers themselves. In spirit, this strategy is consistent with early work on `k-means`, also using centroids to obtain complexity (Inaba et al., 1994) and approximation bounds (Matoušek, 2000). The second one is the reselection strategy, based on the addition rather than the removal of seeds–we never work with a pool of seeds of size $> K$. Overall, our design choices yield reselection schemes which outperform the recently developed multi-swap strategy from (Grunau et al., 2023), (Fig. 1), especially when it comes to running times.

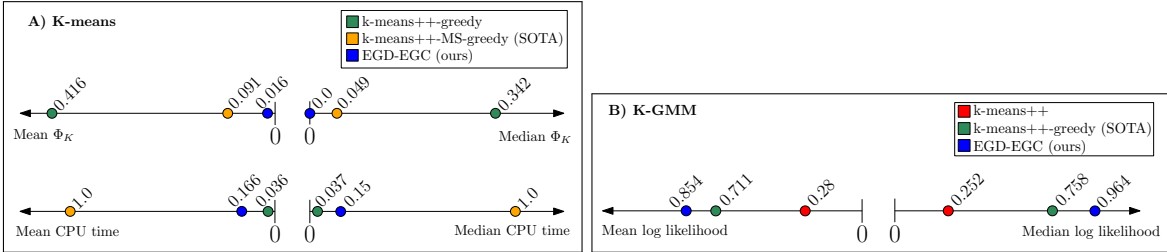

Figure 1: **Gains yielded by our seeding methods. (A, `k-means`)** Mean and median (over 18 datasets) of min-maxed SSE $\Phi_K$ ($m_3(\overline{\Phi}_K)$, see Sec. 6.2) and CPU time ($m_3(\overline{t})$, see Sec. 6.2), the smaller the better–see also Sec. 6. **(B, `k-GMM`)** Mean and median of min-maxed Log likelihood (over 1800 datasets), the larger the better–see also Sec. 7. Seeding methods have negligible impact on CPU time for `k-GMM`.

## 2 Previous work on `k-means`

### 2.1 `k-means`

**`k-means` and its complexity.** In `k-means`, let $c_i$ be the center of mass (COM) of the $i$-th cluster $C_i$. The Sum of Squared Errors (SSE) functional reads as

$$\Phi_K = \sum_{i=1,\ldots,K} \sum_{x_j \in C_i} \|x_j - c_i\|^2 . \tag{1}$$

From the geometric standpoint, if one assumes that $d$ and $K$ are fixed, `k-means` is solvable in $O(n^{O(dk)})$ polynomial time (Inaba et al., 1994). However, if $k$ or $d$ are functions on $n$, `k-means` is NP-hard (Dasgupta, 2008; Mahajan et al., 2012).

From a practical standpoint, so-called *Lloyd* iterations are used to improve an initial set of seeds (Lloyd, 1982), by iterating two steps: (i) ascribe each data point to its nearest center, (ii) recompute the center of mass of each cluster. The process halts when the clusters are stable. The outcome naturally depends on the initial choice of seeds–it is a random variable–and no information is provided with respect to the optimal value of $\Phi_K$, denoted $\Phi_{K,OPT}$.

**Randomized seeding with `k-means++`.** A landmark has been the design of the k-means++ *smart* seeding strategy, which consists of ensuring that the initial centers are correctly placed in the unknown clusters (Arthur & Vassilvitskii, 2007). Assume a set of seeds $S_k$ has been selected, and for each sample in $x \in X \backslash S_k$, let $D^2(x)$ be the square of the minimum distance to a seed. The next seed $c_{k+1}$ is chosen at random from $X \backslash S_k$ using the probability $D^2(\cdot)$. Under this selection scheme, the outcome $\Phi_K$ is a random variable satisfying $\mathbb{E}\left[\Phi_K\right]/\Phi_{K,OPT} \leq 8(\ln K + 2)$ (Arthur & Vassilvitskii, 2007). A useful heuristic also described in (Arthur & Vassilvitskii, 2007) consists of choosing each new seed as the best out of a pool of size $l$. This seeding variant, referred to as *greedy* `k-means++` or `k-means++-G` (Celebi et al., 2013), is implemented in `scikit-learn` with $l = 2 + \log K$ candidates. The method has approximation factors of $O(l^3 \log^3 K)$ and $\Omega(l^3 \log^3 K / \log^2(l \log K))$ (Bhattacharya et al., 2020; Grunau et al., 2023). It is theoretically preferable to use a single seed, as increasing the pool size reduces randomization whence the quality of seeds.

**Improved seeding with reselectors.** So-called *local searches* (LS) consist of replacing seeds by samples when $\Phi_K$ decreases (Kanungo et al., 2002). To replace one seed, the `k-means++-LS` algorithm samples the new candidate seed with the $D^2$ strategy instead of checking all possible options (Lattanzi & Sohler, 2019). Running $Z = \Omega(K \log \log K)$ iterations yields a constant approximation factor (CFA) of 509 (Lattanzi & Sohler, 2019), later improved to $\sim 26.64$ (Grunau et al., 2023). The number of iterations to obtain such a CFA has also been studied (Choo et al., 2020), as well as the complexity of the method (Fan et al., 2023). In practice, though, `k-means++-G` outperforms these improvements–see our Experiments. Last but not least, the *multi-swap* variant consists of opting out $p > 1$ seeds instead of one (Beretta et al., 2023), yielding the `k-means++-MS` algorithm. Using $p = O(1)$ and $Z = O(ndK^{p-1})$ iterations yields a CFA $< 10.48$. Practically, exploring the $\binom{K+p}{p}$ candidate swaps is not effective. Starting from $K+p$ seeds, a greedy variant iteratively discards the seed minimizing the cost increase–thus less representative of the data. This greedy version, denoted `k-means++-MS-G`, outperforms `k-means++-G` in practice (Beretta et al., 2023). But as we shall see, it is outperformed by our seeders, especially for the running time.

**Deterministic seeding.** Deterministic seeding methods have also been proposed, in particular `k-means-var-pca` and `k-means-var-ca` (Su & Dy, 2007). However, they require costly operations, and `k-means++` often performs on par with them.

A thorough experimental comparison has been presented in (Celebi et al., 2013), using 32 datasets up to $\sim 2$ M points and dimension up to $d = 617$. Three methods consistently outperform the remaining ones (Fig. S1): `k-means++`, `k-means++-G`, `k-means-var-pca`.

## 3    Previous work on Gaussian Mixture fitting using EM

Gaussian mixtures models (GMM) are of fundamental interest, both in theory and in practice. We briefly review below recent results on the learnability of GMM and the role of seeding.

**Learnability and connection to seeding.**   A `k-GMM` model is defined by as a weighted sum of Gaussian distributions, that is $\mathcal{N}(x \mid \Theta) = \sum_{k=1}^{K} w_k \mathcal{N}(x \mid \mu_k, \Sigma_k)$, with $\sum_k w_k = 1$. The learnability of GMM received a considerable attention, and we only mention the most recent papers we are aware of, in two veins. The first vein deals with the computation of a GMM close to the unknown one in total variation (TV) distance. Assuming a lower bound on the mixing weights and also on the pairwise TV distance between the components, a probabilistic polynomial time is possible (Liu & Moitra, 2023). It exploits certain algebraic properties of the higher order moments of the Gaussians. But as far as we know, such approaches are not of practical interest. The second vein is the learnability via the recovery of clustering labels; that is, assuming that the samples have been generated by a GMM, one wishes to identify which Gaussian generated which sample. Optimal clustering rates were recently reported (Chen & Zhang, 2024), based on separability hypothesis on the components involving the Mahalanobis distance between the centers. The algorithm uses a hard EM starting from a *decent* initialization, namely a classifier with sublinear loss. Such a warm start is achieved using the vanilla Lloyd algorithm–see also (Gao & Zhang, 2022), and our methods are of direct interest for this step.

**Initialization of EM for GMM fitting.**   When a GMM is fitted using EM, be it soft EM (Bishop & Nasrabadi, 2006) or hard EM (Chen & Zhang, 2024), the outcome depends on the initial mixture, whose design received a significant attention–see (Kwedlo, 2013; Blömer & Bujna, 2016; You et al., 2023) and references therein. In short, the reference methods obtain a partition of the dataset and use the points of the corresponding clusters to estimate the mixture components passed to EM. An interesting observation is that it is often beneficial to estimate isotropic initial components instead of anisotropic ones (Algorithms `Means2SphGMM` and `Means2GMM`), Algo. 2 and (Blömer & Bujna, 2016). Intuitively, along the greedy selection process, the parameters estimated at step $k-1$ are only a very coarse estimate of the $(k-1)$-components of an optimal `k-GMM`.

The initial clustering can be obtained using `k-means++`, yielding the initialization `K-GMM-seeding-++`. However and as recalled in Introduction, since a `k-GMM` is to be estimated, it is interesting to replace the Euclidean distance by alternative better suited to Gaussian components. Of particular interest is the seeding method from (Kwedlo, 2013), as the center of a component is iteratively chosen to maximize the Mahalanobis distance to the already chosen center. Two generalization were proposed in (Blömer & Bujna, 2016):

• The `Spherical Gonzalez (SG)` method chooses a seed by maximizing the Mahalanobis distance to the components already chosen. Doing so faces the risk of choosing outliers, so that the method samples candidates in $S \subset X$, with $|S| = \lceil s|X| \rceil$ – with $s$ a hyperparameter $\in (0, 1]$.

• The `Adaptive (Ad)` method chooses points using a strategy similar to $D^2$, except that the probability distribution used mixes an $\alpha$ component of the Mahalanobis distance, and a $1 - \alpha$ fraction of the uniform distribution on points. This latter component aims at avoiding outliers.

A further independent option has been studied. As noted above, at each step, `Means2SphGMM` is used to define the $K$ mixture components. In addition, one classification EM step (CEM) may be used to refine the mixture (Celeux & Govaert, 1992). (CEM can be seen as a classification version of EM, as it imposes a hard classification step between the E-step and the M-step of EM.)

## 4    New seeding strategies for `k-means`

### 4.1    Selected useful observations

**Notations for Sum of Squared Errors.**   The following notations will be useful:

• $\Phi_K$: the `k-means` SSE functional upon termination of `k-means`– Eq. (1).

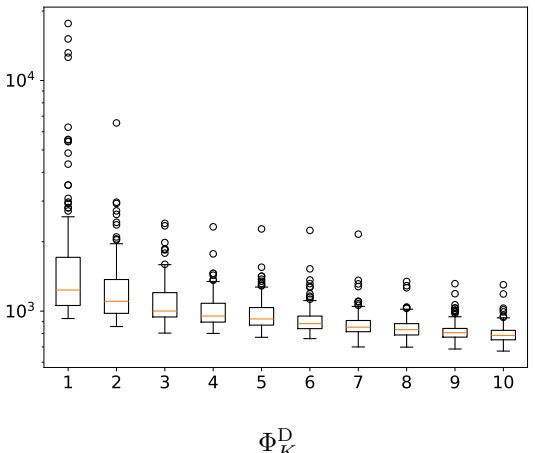 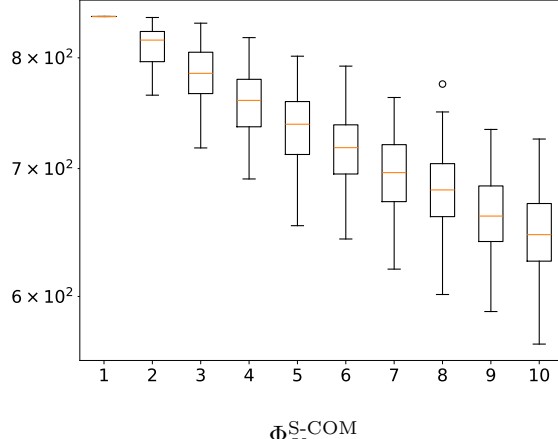

$$\Phi_K^{\mathrm{D}} \qquad\qquad \Phi_K^{\mathrm{S\text{-}COM}}$$

Figure 2: `k-means`: **boxplot of the** $\Phi_K^{\mathbf{D}}$ **and** $\Phi_K^{\mathbf{S\text{-}COM}}$ **values along the seeding selection process for each** $k \in 1, \dots, K$**.** Statistics over 150 repeats on *spam* dataset.

●$\Phi_K^{\mathrm{D}}$: the SSE of data points, using for each such point the distance to its nearest seed–which is also a data point. That is, Eq. (1), where the centers are the $K$ seeds.

●$\Phi_K^{\mathrm{S\text{-}COM}}$: the SSE of data points, using for each such point the distance to the center of mass (COM) of all samples sharing the same seed. That is, Eq. (1), where the centers are the COM of the $K$ clusters associated with $K$ seeds selected.

$D^2$ **distances during seed selection.** Another interesting parameter is the stability of the distribution of distances used by the $D^2$ strategy. The typical behavior is a decreasing variance of the mean (over all points) squared distance $\overline{D}^2$ along the seed selection (Fig. S3). This indicates that distance-wise, the choice of seeds with a large index $k \in 1, \dots, K$ is more stable along seed selections than that of seeds with low index.

$\Phi_K^{\mathbf{D}}$ **and** $\Phi_K^{\mathbf{S\text{-}COM}}$ **during seed selection.** We replicate the analysis just carried out for $\overline{D}^2$ to $\Phi_K^{\mathrm{D}}$ and $\Phi_K^{\mathrm{S\text{-}COM}}$ (Fig. 2 and Fig. S4). $\Phi_K^{\mathrm{D}}$ has a behavior similar to that of $\overline{D}^2$. This suggests that `k-means++-G` suffers from the limitation seen with `k-means++`, as the metric used to compare candidate seeds also stabilizes when the index $k$ increases. Interestingly, statistics for $\Phi_K^{\mathrm{S\text{-}COM}}$ are much more stable along seed selections, with fewer outliers and more concentrated boxes.

## 4.2 Notations for seeding variants

The seeding in `k-means++` and `k-means++-G` selects $K$ seeds in one pass using distances from data points to already selected seeds. We propose a multipass seed selection strategy, each pass being qualified by three ingredients:

●*(Options: sampling candidate seeds)* E: Euclidean distance. (NB: used for the sake of coherence with seeding methods for `k-GMM`, see Section 5.)

●*(Options: size of the pool of candidate seeds)* O: One | G: Greedy. The symbol O (resp. G) refers to a seed selection using a single (resp. $\log K + 2$) candidates–this latter number being that used in the `scikit-learn` implementation.

●*(Options: ranking candidate seeds)* D: Data | C: COM | N: NA. The letter D (resp. C) refers to a selection using the distance between data points (resp. from data points to the centers of masses induced by the seeds). For a pool of size one, there is no such design strategy – whence N or Not Applicable.

Summarizing, a seed selection process is described by the following regular expression:

$$\text{Seeding-}\{\text{E } [\text{O}|\text{G}] \ [\text{D}|\text{C}|\text{N}] \ \}^{+}. \tag{2}$$

Let us illustrate these conventions with `k-means++` and `k-means++-G`:

• `Seeding-EON`: the seeding in `k-means++`. Seeds are selected in one pass; the selection of a seed uses a single candidate (letter O, which implies letter N).

• `Seeding-EGD`: seeding used in `k-means++-G`. Seeds selected in one pass; the seed is selected amidst a pool of candidates (letter G); in this pool, the seed retained is that yielding the lowest $\Phi_K^{\mathrm{D}}$ (Arthur & Vassilvitskii, 2007), which uses the distance from samples to seeds, which are themselves samples (whence the letter D).

### 4.3 New iterative seeding strategies

**Variance reduction:** `Seeding-EGD-EGD`. `k-means++` uses a single pass strategy. To mitigate the influence of the initial steps, we propose a two-pass zig-zag selection process of seeds (Algorithm 1). We have seen that the variance of $D^2$ distances along seed selections decreases during the seed selection (Fig. S3). Therefore, by reselecting the centers a second time, the $D^2$ selection is less dependent on randomness and thus more accurate. This second selection can be done in two ways, by processing the seeds upstream (zig pass, from 1 to $K$), or downstream (zag pass, from $K$ to 1). Experiments have not shown any significant difference, so that we use the downstream order in the sequel of this work. Combining this multi-pass strategy with a greedy selection results in the `Seeding-EGD-EGD` method : the second EGD zag pass reselects seeds, choosing the best amidst a pool of $\log K + 3$ candidates ($\log K + 2$ candidates as in `k-means++-G`, and the center obtained during the zig pass).

**Look-ahead:** `Seeding-EGD-EGC`. In anticipation for LLoyd iterations, we propose to use $\Phi_K^{\mathrm{S\text{-}COM}}$ instead of $\Phi_K^{\mathrm{D}}$ to rank the seeds in a pool of candidates. Indeed, the first Lloyd iteration replaces the initial seeds by the center of mass of that cluster.

In practice, this strategy is only effective for large values of $k \in 1, \ldots, K$, as $\Phi_K^{\mathrm{S\text{-}COM}}$ does not sufficiently discriminate candidates on the first seeds. The extreme case is that of the first seed selected, for which the center of mass is unique. Experiments confirmed this behavior, so that we stick to the seed variant `Seeding-EGD-EGC`, in which we replace $\Phi_K^{\mathrm{D}}$ with $\Phi_K^{\mathrm{S\text{-}COM}}$ during the zag pass.

### 4.4 Seeding and final objective: correlation?

Given that we aim at optimizing the SSE functional $\Phi_K$, the look ahead principle just outlined seems rational. However, it is also instrumental to think about the `k-means` problem in terms of energy / fitness landscape (Dicks & Wales, 2022). To do so, define the *fitness landscape* of a `k-means` problem as set of pairs $\{(\Pi_i, \Phi_{K,i})\}_{i \geq 1}$ obtained during the Lloyd iterations, with $\Pi_i$ the partition / clustering of the point cloud, and $\Phi_{K,i}$ the corresponding SSE. (NB: index 0 corresponds to the seeding outcome.) Let us, intuitively, define a *funneled* fitness landscape as a `k-means` problem such that the sequence of partitions $\Pi_i$ visited during Lloyd iterations eventually leads to a *low* lying local minimum $\Phi_{K,\text{final}}$, that is a clustering whose SSE is close to the optimal value $\Phi_{K,OPT}$. In that case, different initialization with drastically different $\Phi_{K,0}$ may yield to the same value $\Phi_{K,\text{final}}$. Which means that no correlation will be observed between $\Phi_{K,0}$ and $\Phi_{K,\text{final}}$.

To substantiate this intuition, we study the (Pearson and Spearman) correlations $(\Phi_K, \Phi_K^{\mathrm{D}})$ and $(\Phi_K, \Phi_K^{\mathrm{S\text{-}COM}})$ on classical datasets (Celebi et al., 2013), using `k-means++-G` for $\Phi_K^{\mathrm{D}}$ and `Seeding-EGD-EGC` for $\Phi_K^{\mathrm{S\text{-}COM}}$ (Fig. S2). A mild correlation is observed is all cases, confirming our expectations. Importantly, this fact does not contradict the approximation factors discussed in previous work: the approximation factors qualify the distance to the optimal SSE $\Phi_K$, while the aforementioned correlations depend on the topography of the `k-means` fitness landscape, or, phrased differently, in the sequence of partitions $\Pi_i$ visited during Lloyd iterations.

# 5 New seeding strategies for `k-GMM`

## 5.1 Notations for seeding variants

The seeding methods developed for `k-GMM` follow the multipass strategy introduced for `k-means`. Yet, the use of a mixture model allows for new metrics both in the sampling and selection of candidate seeds. Thus, our new seeding strategies consists in combinations of `k-GMM` seeding passes, followed by the use of `Means2GMM` to transform the selected seeds into an initial model to be optimized. (NB: the seeds are used as $\mu$s arguments in the Algorithm 2.) The `k-GMM` passes are qualified with three ingredients:

•*(Options: sampling candidate seeds)* E: Euclidean distance | A: Adaptive Mahalanobis | G: Gaussian distance. The symbol E corresponds to the Euclidean distance originally used for $D^2$ weighting in `k-means++`. The symbols A refers to the distances used in the Adaptive seeding method (with $\alpha = 0.5$) (Blömer & Bujna, 2016). Finally, G refers to a strategy using the $D^2$ method on distances (Eq. 13) between Gaussian distributions estimated at every data point. See details in Section S1.3.

•*(Options: size of the pool of candidate seeds)* O: One | G:Greedy. Similar to `k-means`.

•*(Options: ranking candidate seeds)* D: Data | C: COM | L: Log-likelihood | N: NA. The letters D, C and N match the options used for `k-means`. We add a new metric with the letter L, corresponding to the log-likelihoods of mixture models estimated using each candidate seed.

Summarizing, a seed selection process is described by the following regular expression:

$$\text{K-GMM-seeding-}\{[\text{E}|\text{A}|\text{G}] \ [\text{O}|\text{G}] \ [\text{D}|\text{C}|\text{L}|\text{N}] \ \}^+. \tag{3}$$

## 5.2 New iterative seeding strategies

### 5.2.1 Greedy adaptive selection: `K-GMM-seeding-AGL`

The adaptive selection strategy combines Mahalanobis and uniform distances to select seeds (Blömer & Bujna, 2016), before running `Means2SphGMM` to obtain the GMM passed to EM. The seeding `K-GMM-seeding-AGL` adds to this strategy a selection based on the likelihood, a look-ahead with respect to the EM steps.

### 5.2.2 EM look-ahead : `K-GMM-seeding-EGD-EGL`

In a look-ahead spirit similar to that introduced with $\Phi_K^{\text{S-COM}}$ for `k-means`, we use the log-likelihood as a selection metric to rank seeds among candidates. To obtain a log-likelihood value from a set of seeds, the `Means2GMM` algorithm is used to construct a temporary Gaussian mixture model (Blömer & Bujna, 2016). Similarly to the use of $\Phi_K^{\text{S-COM}}$ for `k-means`, the log-likelihood is irrelevant to discriminate candidates for small values of $k$. Consequently, the `K-GMM-seeding-EGD-EGL` seeding variant uses it for the zag pass only.
*Remark* 1. The observations raised for `k-means` (Sec. 4.4) are also valid for `k-GMM`.

### 5.2.3 EM look ahead and adaptive selection in the zag pass: `K-GMM-seeding-EGD-AGL`

We combine the best features used in the previous two methods. First, the adaptive selection using a GMM, as in `K-GMM-seeding-AGL`. The adaptive selection is restricted to the second pass though, to ensure that the GMM is representative of the cluster structure of the data. Second, the two pass and look-ahead strategy of `K-GMM-seeding-EGD-EGL`. The resulting method is called `K-GMM-seeding-EGD-AGL`.

### 5.2.4 $D_G^2$ with Gaussian distance

Finally, we explore the use of `k-means++-G` based on a distance between Gaussians locally estimated at each sample, see method `K-GMM-seeding-GGD` in Section S1.3.

# 6  `k-means` **seeding: tests**

## 6.1  Experimental protocol

**Implementation and stop criterion.** We compare our seeding methods with `k-means++-LS` (with $Z = k$) and `k-means++-MS-G` (with $Z = K$, $p = 2 + log(K)$) to initialize `k-means`. We chose $Z = k$ to match the number of swaps performed by our methods, and $p = 2 + log(K)$ to match the candidate pool size originally proposed for greedy kmeans++ in (Arthur & Vassilvitskii, 2007). All methods were implemented in C++ using the Eigen library. The Lloyd iterations are stopped when the Frobenius norm of the difference in the center clusters is smaller than $1e-4$. As a failsafe, the Lloyd iterations are also stopped after reaching a maximum number of iterations of 50. A python implementation of the best contender, `Seeding-EGD-EGC`, is available from `https://gitlab.inria.fr/abs/improved-seeding-kmeans-kgmm`. An optimized C++ implementation is also available in the package SBL::Cluster_engines of the Structural Bioinformatics Library `https://sbl.inria.fr/doc/Cluster_engines-user-manual.html`.

**Datasets.** Our experiments involve 12 datasets from the UCI Machine Learning Repository, 11 of which are a subset of the 32 used in (Celebi et al., 2013): {*Cloud cover, Corel image features, Steel plates faults, Letter recognition, Multiple features, Musk (Clean2), Optical digits, Pen digits, Image segmentation, Shuttle (Statlog), Spambase, Yeast*} [1].

These 12 datasets are the most challenging ones, due to variability incurred by the seeding strategies, and its effect on the final clustering (Celebi et al., 2013). The value of $K$ for a dataset is that provided alongside each dataset. They range in size from 1024 to 58,000 data points.

We also process the {KDD-BIO, KDD-PHY, RNA} datasets from (Lattanzi & Sohler, 2019) used in the assessment of the SOTA method `k-means++-MS-G` (Beretta et al., 2023). They range in size from 100k to 485k points. Following (Lattanzi & Sohler, 2019), we cluster them with $K = \{25, 50\}$.

In total, we investigate $12 + 3 + 3 = 18$ datasets. Following common practice, on a per dataset basis, we perform a min-max normalization on the coordinates to avoid overly large ranges.

**Hardware.** Calculations were run an a HP desktop computer running Fedora Core 39, equipped with 24 CPUs (i9-13900K) and 131 GB or RAM.

## 6.2  Statistical assessment

**Chosen statistics.** To compare the various methods, we consider the SSE $\Phi_K$ upon convergence of the Lloyd iterations. We also measure the CPU time of each seeding strategy, as well as the average number of Lloyd iterations needed for convergence in the following `k-means`. As the seeding methods are non-deterministic, we report average results over a set of $R = 100$ repeats. For each dataset, we also provide p-values to assess the null hypothesis stating two distributions of observed values–that of the best performing method and that of a contender–are identical. We use two non parametric two-sample tests: first, the Mann-Whitney U-test (U) whose test statistic uses the ranks of values; second the Kolmogorov-Smirnov (KS) test, which uses the difference between the cumulative distribution functions. We use these two tests to chase statistical power, since the U test overlooks the numerical values, and the KS test compares CDF while we care for performances only.

**Comparison across datasets and normalization issues.** To compensate the variability of statistics across datasets, we min-max normalize the statistics of interest ($\Phi_K$ and running time $t$) in two ways. Let $m \in \mathcal{M}$ the particular method to be assessed and $\mathcal{R}$ the set of repeats of that method on a dataset. To compare methods, the first normalization uses the average values (computed over repeats), assigning the value 0 (resp. 1) to the worst (resp. best) method on a dataset – with $m_3(\cdot)$ standing for min-max-Mean:

$$m_3(\overline{\Phi}_{K,m}) = \frac{\overline{\Phi}_{K,m} - \min_{m' \in \mathcal{M}} \overline{\Phi}_{K,m'}}{\max_{m' \in \mathcal{M}} \overline{\Phi}_{K,m'} - \min_{m' \in \mathcal{M}} \overline{\Phi}_{K,m'}} \tag{4}$$

---

[1] *Spambase* is the one dataset not used in (Celebi et al., 2013)

To compare the methods on the range of possible values observed for a given dataset, the second normalization reads as – with $m_G(\cdot)$ standing for min-max-Global:

$$m_G(\overline{\Phi}_{K,m}) = \frac{\overline{\Phi}_{K,m} - \min_{m' \in \mathcal{M}, r \in \mathcal{R}} \Phi_{K,m',r}}{\max_{m' \in \mathcal{M}, r \in \mathcal{R}} \Phi_{K,m',r} - \min_{m' \in \mathcal{M}, r \in \mathcal{R}} \Phi_{K,m',r}} \tag{5}$$

For the running times, we define likewise $m_3(\bar{t}_m)$.

### 6.3 Results

**Incidence of the seeding on $\Phi_K$.** We study the contenders from Section 4. To assess the changes brought by the zig-zag strategy itself, we involve `Seeding-EGDx2`, namely `k-means++-G` where we double the amount of candidates.

Three observations stand out :

●*Best method.* In terms of $m_3(\overline{\Phi}_K)$, `Seeding-EGD-EGC` outperforms all contenders but `k-means++-MS-G` on datasets where seeding has a significant impact (Fig. 3, Fig. 4, Table 1, SI Table S1).

The comparison `Seeding-EGD-EGC` versus `k-means++-MS-G` shows that these methods perform on par for $m_3(\overline{\Phi}_K)$, yet with much better running times for `Seeding-EGD-EGC`– see below.

●*The zag pass is beneficial when using greedy strategies.* The $D^2$ weighting coupled to the greedy seed selection significantly benefits from the zag pass. As shown with the consistent improvements on several datasets observed with `Seeding-EGD-EGD` and `Seeding-EGD-EGC` (Fig. 3, Fig. 4, Table 1, SI Table S1). Specifically, the zig-zag methods outperform `Seeding-EGDx2` on most datasets, providing better seeds while considering the same amount of candidates.

●*Using center of masses to select seeds yields superior results.* The method `Seeding-EGD-EGC` outperforms `Seeding-EGD-EGD` (Fig. 3, Fig. 4, Table 1, SI Table S1) confirming that $\Phi_K^{\text{S-COM}}$ is a better fit than $\Phi_K$.

**Running time.** We study in tandem the CPU total times (Fig. 5, SI Table S2) and the number of Lloyd iterations to reach convergence (Fig. S5).

●*Zig-zag seeding methods are slower than single pass equivalents but reduce the number of Lloyd iterations.*

Let us first compare `Seeding-EGD-EGC` and `k-means++-G`: focusing on the initialization step only, our method incurs an increase of running time by $\sim \times 2$-$3$; taking the entire execution into account (seeding +Lloyd iterations) the increase in running time is around $\sim \times 1.5$ (Fig. 5, Table S2).

Similarly, let us compare `Seeding-EGD-EGC` and the SOTA method `k-means++-MS-G`: for the initialization running time, `k-means++-MS-G` is $\sim \times 6$ times slower than `Seeding-EGD-EGC`; for the total running time, `k-means++-MS-G` is $\sim \times 4$ times slower (Fig. 5, Table S2). This owes to the following asymmetry: our reselection requires finding the best seed to add (from a pool of $l$ candidates) once one seed has been removed; the greedy multi-swap requires finding the best seed to remove after adding $p$ of them–from a pool of size $K + p$ (first seed removed) to $K + 1$ (last seed removed). Moreover, without any optimization / additional storage, the $\Phi_K$ cost update is $O(n)$ for an addition, and $O(Kn)$ for a removal.

●*Seeding methods producing lower values of $\Phi_K$ also tend to reduce the number of required Lloyd iterations.* We observe an average Pearson correlation coefficient of $0.816$ between $\Phi_K$ and the number of Lloyd iterations (Fig. S5). This is expected, as efficient seeding precisely aims at placing seeds near the optimal positions, reducing the number of Lloyd iterations needed for to reach these positions.

### 6.4 Sensitivity to the pool size

While our experiments were conducted with $\log K + 2$ candidates–as in the standard the `scikit-learn` implementation, it naturally makes sense to study the sensitivity of the methods to the pool size $l$. To this end, we consider three pool sizes on different scales, namely $l \in \{2 + logK, 2 + \sqrt{K}, \max(2, K)\}$, reporting

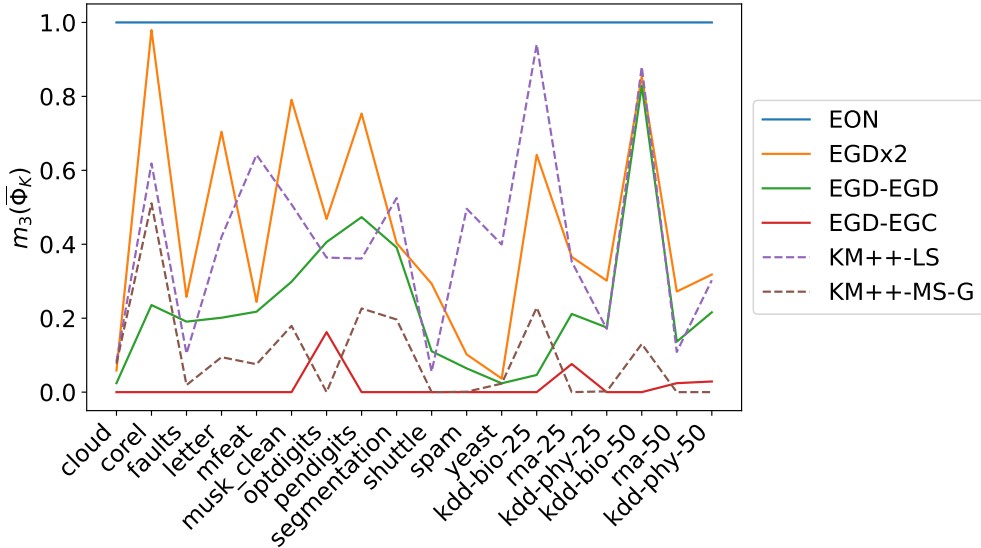

Figure 3: `k-means`: **min-max normalized value** $m_3(\overline{\Phi}_K) -$ **Eq.** (4)**, as a function of the seeding method.** For the reference: the seeding used in `k-means++-G` is `Seeding-EGD`, and `Seeding-EGDx2` is the same with twice as many seeds to match the zig-zag strategy.

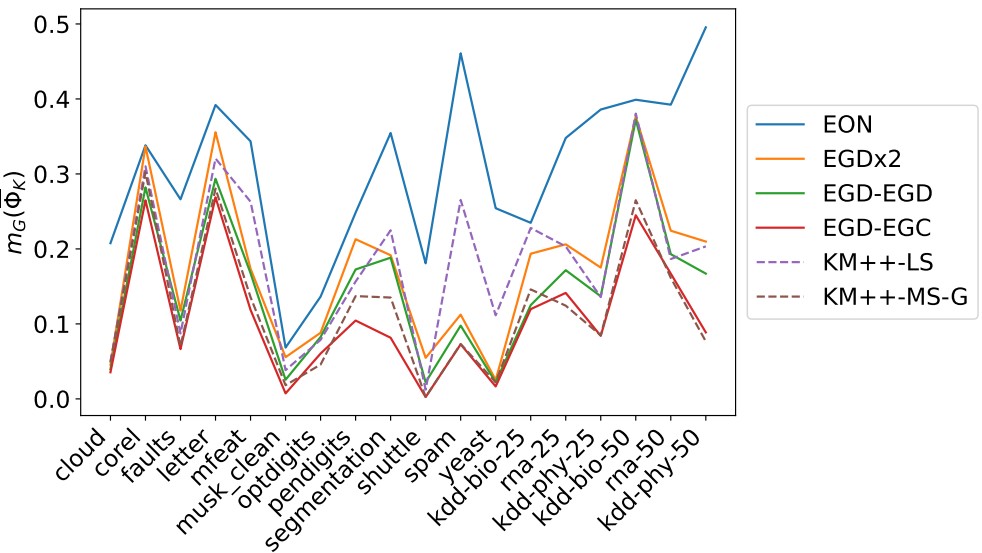

Figure 4: `k-means`: **min-max normalized** $m_G(\overline{\Phi}_K) -$ **Eq.** (5) **as a function of the seeding method.**

| EGD-EGC vs. | cloud | corel | faults | letter | mfeat | musk_clean | optdigits | pendigits | segmentation | shuttle | spam | yeast | kdd-bio-25 | rna-25 | kdd-phy-25 | kdd-bio-50 | rna-50 | kdd-phy-50 |
|---|---|---|---|---|---|---|---|---|---|---|---|---|---|---|---|---|---|---|
| EON (U-test) | 0.000 | 0.006 | 0.000 | 0.000 | 0.000 | 0.000 | 0.000 | 0.000 | 0.000 | 0.000 | 0.000 | 0.000 | 0.017 | 0.000 | 0.000 | 0.000 | 0.000 | 0.000 |
| EON (K-test) | 0.000 | 0.054 | 0.000 | 0.000 | 0.000 | 0.000 | 0.010 | 0.000 | 0.000 | 0.000 | 0.000 | 0.000 | 0.036 | 0.000 | 0.000 | 0.000 | 0.000 | 0.000 |
| EGDx2 (U-test) | 0.578 | 0.013 | 0.006 | 0.001 | 0.007 | 0.000 | 0.041 | 0.000 | 0.000 | 0.768 | 0.001 | 0.003 | 0.123 | 0.000 | 0.000 | 0.000 | 0.000 | 0.000 |
| EGDx2 (K-test) | 0.908 | 0.078 | 0.004 | 0.016 | 0.010 | 0.016 | 0.054 | 0.000 | 0.000 | 0.054 | 0.010 | 0.000 | 0.470 | 0.002 | 0.000 | 0.000 | 0.000 | 0.000 |
| EGD-EGD (U-test) | 0.832 | 0.782 | 0.007 | 0.314 | 0.011 | 0.014 | 0.180 | 0.000 | 0.000 | 0.192 | 0.062 | 0.101 | 0.431 | 0.057 | 0.000 | 0.000 | 0.029 | 0.000 |
| EGD-EGD (K-test) | 0.908 | 0.908 | 0.024 | 0.470 | 0.111 | 0.702 | 0.368 | 0.000 | 0.000 | 0.368 | 0.282 | 0.036 | 0.815 | 0.054 | 0.000 | 0.000 | 0.036 | 0.000 |
| KM++-MS-G (U-test) | 0.202 | 0.276 | 0.865 | 0.557 | 0.686 | 0.098 | 0.076 | 0.027 | 0.071 | 0.088 | 0.572 | 0.038 | 0.772 | 0.303 | 0.515 | 0.813 | 0.342 | 0.051 |
| KM++-MS-G (K-test) | 0.583 | 0.702 | 0.968 | 0.702 | 0.815 | 0.994 | 0.111 | 0.001 | 0.004 | 0.470 | 0.583 | 0.002 | 0.702 | 0.470 | 0.702 | 0.470 | 0.470 | 0.054 |

Table 1: `k-means`: **p-values for Mann-Whitney U-tests and Kolmogorov-Smirnov K-tests between EGD-EGC and each method, on each dataset.** Low p-values indicate that the difference in results is statistically significant. Three color level are used: $[0, 0.05)$: green; $[0.05, 0.15)$: pale green; $[0.15, 1]$: white.

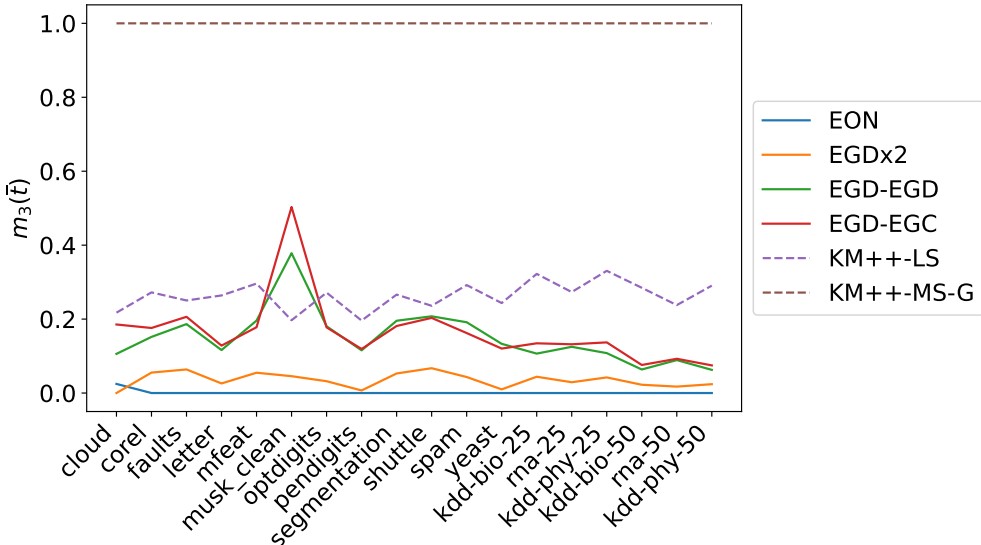

Figure 5: `k-means`: **min-max normalized CPU total time** $m_3(\bar{t})$ **for each seeding method.**

the min-max normalized values ($m_3(\overline{\Phi}_K)$, Fig. S6; $m_G(\overline{\Phi}_K)$, Fig. S7) and CPU times ($m_3(\bar{t})$, Fig. S8; raw times, Fig. S9).

This comparison calls for two comments. First, the larger the pool size the better the result (and the longer the running time). This is somewhat expected for selected datasets, keeping in mind however that for others, increasing the pool size tames down randomization, yielding a worse approximation factor (Bhattacharya et al., 2020; Grunau et al., 2023).

Second, the span observed for $\Phi_K$ decreases when moving from `k-means++-G` to `Seeding-EGD-EGC`. The lesser sensitivity of our method to the pool size illustrates its ability to identify meaningful seeds at the onset, a pattern shared by `k-means++-MS-G`.

## 7 `k-GMM` **seeding: tests**

### 7.1 Experimental protocol

**Implementation and stop criterion.** All seeding methods as well as the EM iterations were implemented in C++ using the Eigen library. Implementation to be released upon publication of the paper.

The stopping criterion for the EM iterations targets the relative difference in log-likelihood between two iterations, and reads as $|l(\theta_n)-l(\theta_{n-1})|/|l(\theta_{n-1})| < 1e-4$. Alternatively, the EM iterations are stopped after reaching a maximum number of 100 iterations.

**Generated datasets.** Following (Blömer & Bujna, 2013), we use GMMs defined from the following parameters: (i) the separation between components (values $s = 0.5, 1, 2$), (ii) the weights of components (uniform, different), (iii) the size of components (constant, different), and (iv) their eccentricity ($e = \max_d \lambda_d/\min_d \lambda_d$; $e \in [1, 2, 5, 10]$). (NB: code available from `https://github.com/mdqyy/simple-gmm-initializations`.) In total, 30 combinations (out of the 48 possible) are selected, and further aggregated into three so-called *groups*: *spherical* (12 models), *elliptical* (9 models), *elliptical-difficult* (9 models). Each GMM is used to generate $D = 30$ datasets, yielding a total of 900 datasets, each involving $n = 10,000$ points.

We also consider *noisy* datasets, namely noisy spherical/elliptical/elliptical-difficult models. To generate a noisy dataset, we generate 9000 points from the noise free model, and add 1000 points drawn uniformly at random in the expanded bounding box of the 9000 samples.

Summarizing, we consider in the sequel six groups involving 60 models (30 noise free, 30 noisy), for a total of $D = 1800$ datasets.

**Grid dataset.** In addition, we include an artificial pathological case with the *grid dataset*. In this group, we include data sampled from a single handcrafted 3-dimensional GMM composed of 27 Gaussians forming the shape of a 3d cubic grid (Fig.S11). The rationale is to highlight situations where the `K-GMM-seeding-GGD` initialization method might be particularly well suited, as the data is generated using a GMM with easily identifiable but highly intersecting and eccentric components. With this GMM, we also generate 30 datasets, each composed of 250 * 27 = 6750 points.

**Statistics.** We aim at comparing $N_c = 9$ initialization contenders using the log-likelihood – Eq. (8), for every group of models out of six. For a given dataset, final log-likelihood values are obtained by averaging the results of $R = 30$ runs of each initialization method in order to assess the variance inherent to their randomness. This yields a vector of final log-likelihood values of size $N_c$ for each dataset (900 vectors for noise free datasets, 900 for noisy datasets). Consider the resulting $D \times N_c$ matrix, with $D = 1800$ and $N_c = 9$. To be able to accumulate results over different datasets from the same group, we perform min-max scaling over each matrix row, such that its entries are in $[0, 1]$ (with 0 (resp. 1) corresponding to the worst (resp. best) performing method at each row). These rescaled values are termed the *min-max normalized log-likelihoods*. The matrix columns can then be split into 6 blocks, each corresponding to a specific group. To compare the six groups, we average the *min-max normalized log-likelihood* values of each method on all datasets of a given block – resulting in $6 \times N_c$ values in total.

## 7.2 Results

**Seeding and the final log-likelihood.** The following observations stand out (Fig. 1, Fig.6, Fig.S12, SI Table S3 (exact values)).

•*The zig-zag strategy is state-of-the-art.* EM combined by the classical `k-means++` seeding, a.k.a `K-GMM-seeding-EGD`, is outperformed by the variant using twice as many candidates ( `K-GMM-seeding-EGD(x2)`), which is itself outperformed by `K-GMM-seeding-EGD-EGC`. Consistent with `k-means`, this corroborates the general efficacy of the zig-zag strategy in increasing the final log-likelihood on all classes of datasets.

•*Log-likelihood (LL) based methods are sensitive to noise.* The comparison between noise free and noisy datasets yields a clear separation between methods using the log-likelihood for seed selection. As illustration comparison is that between `K-GMM-seeding-EGD-EGC` and `K-GMM-seeding-EGD-EGL`– which differ only by the metric used to rank candidates. One the one hand, LL based methods are amongst the best on our noise free datasets. On the other hand, these methods appears highly sensitive to outliers (Fig. S13 and Fig. S14).

•*Seeding using the Gaussian based distance is highly effective for mixtures with intersections.* The case of `K-GMM-seeding-GGD`, which uses the $D^2$ strategy on the Gaussian distance performs on par with `K-GMM-seeding-EGD` for generated datasets. On the grid datasets, it is the best performing method overall, and significantly outperforms `K-GMM-seeding-EGD` as a single pass strategy (Fig. S12). As opposed to the likelihood regulated methods, `K-GMM-seeding-GGD` incorporates the advantages of estimating Gaussian components without suffering from the inclusion of noise.

**Running time.** Similarly to the `k-means` case, the zig-zag seeding methods are slower that their one pass counterpart (Fig. S15), but the SSE regulated methods remain competitive (Fig.S16). Most importantly, the likelihood regulated passes are 5x to 6x slower than the SSE regulated passes. Finally, the `K-GMM-seeding-GGD` method is around two to three orders of magnitude slower than all shown methods due to the cost of estimating Gaussians on each data points.

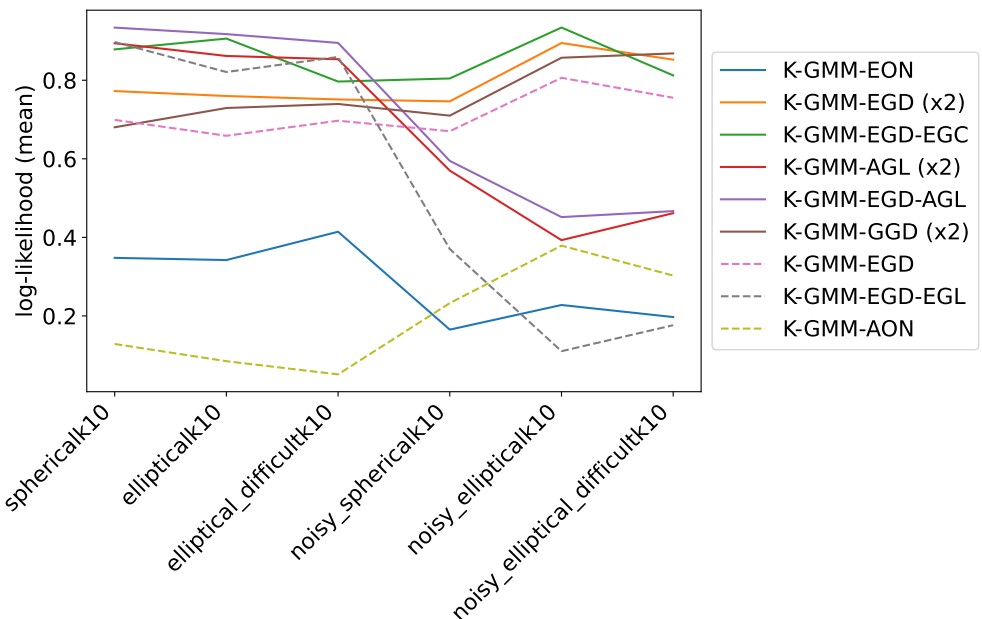

Figure 6: `k-GMM` : **mean of the min-max normalized log-likelihood over datasets of each scenario.** The larger the log-likelihood, the better. See text for details.

## 8 Outlook

Clustering is a fundamental problem, and `k-means` is a fundamental approach to it. In search for efficient and provably correct solutions, the smart seeding approach of `k-means++` play a central role. Re-seeding methods based on local searches and multi-swaps recently underwent important developments, both in theory and in practice. We improve on these in several ways.

On the design side, while recent re-seeding methods have used distances to the data points, we use distances to the centroids of the clusters induced by the centers, in the spirit of the early works of local searches. Also, our methods are particularly simple and do not require any elaborate data structure to maintain nearest neighbors of the seeds under scrutiny during re-selection. Our methods achieve SOTA performance but currently lack theoretical guarantees, specifically in terms of a constant approximation factor (CFA). This situation parallels that of `k-means++-G`, which has been the preferred practical method since 2007 despite the absence of theoretical analysis until 2023, and since 2023 despite having a CFA worse than that of `k-means++`. Our experiments also shed light on subtle properties of `k-means` often overlooked, including the (lack of) correlations between the SSE upon seeding and the final SSE, the variance reduction phenomena observed in iterative seeding methods and the sensitivity of the final SSE to the pool size for greedy methods.

Practically, we anticipate that our best seeding methods will become one of the standard seeding technique(s). However, the analysis of our methods raises challenging questions. The first one is the role of the metric and its coherence with the functional eventually optimized by Lloyd iterations for `k-means`, or EM iterations for `k-GMM` .

The second relates to the ordering along which seeds are being opted out during reselection, as we only used the reversed selection order in this work. For example, in a manner akin to simulated annealing, which lowers the temperature strategy, re-seeding seeds with *higher* impact on variance first may provide finer results. These intuitions have to be formally established and tested.

**Acknowledgements**

This work has been supported by the French government, through the 3IA Côte d'Azur Investments (ANR-19-P3IA-0002), and the ANR project Innuendo (ANR-23-CE45-0019).

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

## S1    Supporting information: theory

### S1.1    `k-means`

---
**Algorithm 1  `Seeding-EON-EON`.**

---
1: **procedure**  `Seeding-EON-EON`$(data, K)$

2:    $centers \leftarrow$  `Seeding-EON` (data, K)
3:    ▷ *Reselect centers in reverse order*                                                             ◁
4:    **for** $k \leftarrow K$ to 1 **do**
5:        Delete $centers[k]$
6:        Choose $new\_c_k$ with the $D^2$ strategy
7:        Insert $new\_c_k$ in $centers$ at position $k$

---

### S1.2    EM for Gaussian Mixtures

#### S1.2.1    The `Means2GMM` and `Means2SphGMM` algorithms

Consider a dataset and a hard partition of this dataset into clusters.

The algorithm 2 (Blömer & Bujna, 2016) converts this partition into a Gaussian mixture model. An interesting observation is that it is often beneficial to estimate isotropic initial components instead of anisotropic ones.

---
**Algorithm 2 The classical `Means2GMM` algorithm.** The variant `Means2SphGMM` consists of changing the full anisotropic estimation of line 7 by the isotropic estimation of line 8. (Blömer & Bujna, 2016).

---
1: **procedure** `Means2GMM`$(X, \mu_1, ..., \mu_K)$

2:        Derive partition $C_1, ..., C_K$ of X by assigning each point $x_i \in X$ to its closest mean
3:        ▷ *Build GMM components*                                                             ◁
4:        **for** $k \leftarrow 1$ to $K$ **do**
5:            $\mu_k = 1/|C_k| \sum_{x \in C_k} x$
6:            $w_k = |C_k| \,/\, |X|$
7:            $\Sigma_k = 1/|C_k| \sum_{x \in C_k} (x - \mu_k)(x - \mu_k)^\mathsf{T}$
8:            If $\Sigma_k$ is not positive definite, take $\Sigma_k = 1/(d|C_k|) \sum_{x \in C_k} \|x - \mu_k\|^2 \mathbf{I}_d)$
9:            If $\Sigma_k$ is still not positive definite, take $\Sigma_k = \mathbf{I}_d$

---

#### S1.2.2    The E and M steps

In the following, we recall the EM algorithm to fit a GMM. The algorithm involves two steps.

**The E-step.**    Given the functions at iteration $t$, one computes the responsibility of the gaussian $g_j^{(t)}$ for the sample point $x_i$

$$r_{ij}^{(t)} = \frac{w_j^{(t)} g_j^{(t)}(x_i|\Theta_j^{(t)})}{\sum_{k=1}^{M} w_k^{(t)} g_k^{(t)}(x_i|\Theta_k^{(t)})}. \tag{6}$$

The sum of responsibilities associated to one component then read as

$$n_j^{(t)} = \sum_{i=1}^{N} r_{ij}^{(t)} \tag{7}$$

It may be noted (Bishop & Nasrabadi, 2006)(Chapter 9) that the weight $w_j^{(t)}$ is the prior probability for the sample $x_i$ to be generated by the j-th component; the responsibility $r_{ij}^{(t)}$ is the corresponding posterior probability.

**The M-step.** Re-estimate the parameters of $g_j^{(t+1)}$ using the maximum likelihood:

$$w_j^{(t+1)} = \frac{n_j^{(t)}}{N},$$

$$\mu_j^{(t+1)} = \frac{1}{n_j^{(t)}} \sum_{i=1}^{n} r_{ij}^{(t)} x_i,$$

$$\Sigma_j^{(t+1)} = \frac{1}{n_j^{(t)}} \sum_{i=1}^{n} r_{ij}^{(t)} (x_i - \mu_j^{(t+1)})(x_i - \mu_j^{(t+1)})^{\mathsf{T}}.$$

**Convergence.** Consider the log likelihood for the $n$ samples, that is

$$\mathcal{LL}(X \mid \Theta) = \ln \mathcal{N}(X \mid \Theta) = \sum_{i=1,\dots,n} \ln \Big( \sum_k w_k \mathcal{N}(x_i \mid \mu_k, \Sigma_k) \Big). \tag{8}$$

One checks the convergence of the mixture parameters, or of the likelihood.

**Numerics.** In the E-step of the EM algorithm, the evaluation of the GMM needed to compute responsibilities induces a risk of underflow. This is due to the evaluation of singular components on datapoints for which they have no responsibility, resulting in pdf values that tend to 0.
To solve this problem, we adapt the E-step by computing the responsibilities using only the logarithmic scale. We first compute the log-pdfs of singular components for individual unnormalized responsibilities. To obtain normalized responsibilities, we must compute the log pdf of the whole mixture model.

This computation requires a summation of the pdf values of components, which cannot be explicitly obtained without losing the logarithmic scale. We avoid this problem by performing the logsumexp trick (https://en.wikipedia.org/wiki/LogSumExp) using the log-pdfs of individual components, allowing us to obtain the log-pdf of the whole mixture model while staying in logarithmic scale throughout.

The logsumexp operation consists in the following:

$$logsumexp(x_1, \dots, x_n) = \log \Big( \sum_{i=1}^{n} \exp(x_i) \Big) \tag{9}$$

Applied on the log-pdfs of individual components, this allows us to obtain the log-pdf of the mixture model, but loses the logarithmic scale. Therefore, we apply the logsumexp trick by using the following equivalent to the logsumexp operation:

$$logsumexp(x_1, \dots, x_n) = x^* + \log \Big( \sum_{i=1}^{n} \exp(x_i - x^*) \Big) \tag{10}$$

This equivalent allows us to shift the values in the exponent by an arbitrary constant. We can then set $x^* = \max\{x_1, \dots, x_n\}$, to ensure the largest exponentiated term is equal to $\exp(0) = 1$, avoiding the risk of underflow on the result of the logsumexp operation.

## S1.3 $D_G^2$ with Gaussian distance

**Multivariate Gaussians and associated distance** The parameter set of a multi-dimensional Gaussian is denoted $\Theta = (\mu, \Sigma)$; it has dimension $d + d(d+1)/2 = d(d+3)/2$. The set of positive semidefinite matrices, to which covariance matrices belong, is denoted $S_d^+$.

The density of a multivariate Gaussian reads as:

$$g(x|\Theta) = \frac{1}{\sqrt{(2\pi)^d |\Sigma|}} \exp(-\frac{1}{2}(x-\mu)^\mathsf{T}\Sigma^{-1}(x-\mu)). \tag{11}$$

Let $\Sigma_1$ and $\Sigma_2$ be two PSD matrices; let $\mu_{12} = \mu_1 - \mu_2$, and let $\Sigma_{12} = (\Sigma_1 + \Sigma_2)/2$. Consider the generalized eigenvalue problem (GEP) $\Sigma_1 V = \lambda \Sigma_2 V$, with $V$ the column matrix of the generalized eigenvectors. The Riemannian metric for PSD reads as (Förstner & Moonen, 2003):

$$d_{PSD}(\Sigma_1, \Sigma_2) = \Big(\sum_{i=1}^{d} \log^2 \lambda_j\Big)^{1/2}. \tag{12}$$

Using this, one defines the following distance between Gaussian distributions (Abou-Moustafa et al., 2010):

$$d_G(g_1, g_2) = \big(\mu_{12}{}^\mathsf{T}\Sigma^{-1}\mu_{12}\big)^{1/2} + \Big(\sum_{k=1}^{d} \ln^2 \lambda_k\Big)^{1/2}. \tag{13}$$

`K-GMM-seeding-GGD`**: details.** Using the previous gaussian distance we propose the `K-GMM-seeding-GGD` seeding algorithm. This is a modification of the `K-GMM-seeding-EGD` algorithm, where the `k-means++` seeding method is used on locally estimated gaussians at each data point, instead of the data points themselves. In other words, this method aims at sampling seeds from the datapoints by considering the shape of the gaussian components that would result from selecting them. This local estimation of gaussians is done as follows:

- At each data point $x_j$, compute average distance $\hat{d}_j$ to $L$-nearest neighbors $\{x_l^{(j)}\}_{l=1}^{L}$ with the following equation. Considering K components will be estimated in the following EM algorithm, we set $L$ to be equal to $N/K$ .

$$\hat{d}_j = \frac{1}{L}\sum_{l=1}^{L} ||x_l^{(j)} - x_j||. \tag{14}$$

- For each pair of points $x_i$ and $x_j$, compute the local distance weighted responsibility of point $x_j$ for point $x_i$ with the following equation. These responsibilities are designed to correspond to the evaluation on the point $x_i$ of an isotropic gaussian with variance $\hat{d}_j$, centered on the data point $x_j$.

$$LG_j(x_i) = \frac{1}{(\hat{d}_j)^D (2\pi)^{D/2}} \exp(-\frac{1}{2}\frac{||x_i - x_j||^2}{(\hat{d}_j)^2})$$
$$\tilde{r}_{ij} = \frac{LG_j(x_i)}{\sum_{k=1}^{N} LG_k(x_i)}.$$

- At each data point, compute local gaussians $G_j$ with the M-step update rules of the EM algorithm (S1.2.2), using the local responsibilities $\tilde{r}_{ij}$.

This process provides a set $G$ of locally estimated gaussians of size $N$, one gaussian for each datapoint. It is then followed by a selection of $K$ gaussians among $G$ with the `k-means++` algorithm using the gaussian distance of Eq.13. Finally, we select the $K$ data points from which the $K$ selected gaussians were obtained as starting centers for the EM iterations, completing the seed selection.

## S2 Supporting information: results

### S2.1 `k-means`

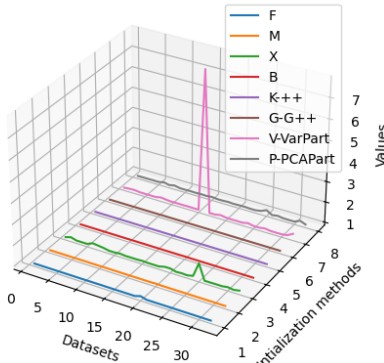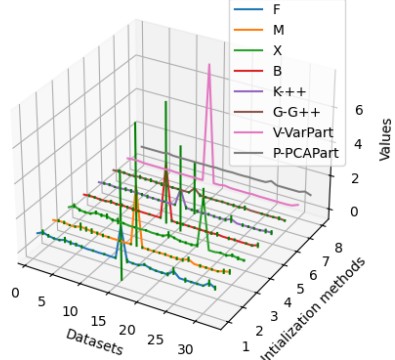

Figure S1: **Plot of Table 2 from (Celebi et al., 2013), on 32 datasets. (Left)** Minimum values scaled by the minimum for a dataset **(Right)** Mean values scaled by the minimum for a dataset. The error bars correspond to mean values $\pm$ the std deviation.

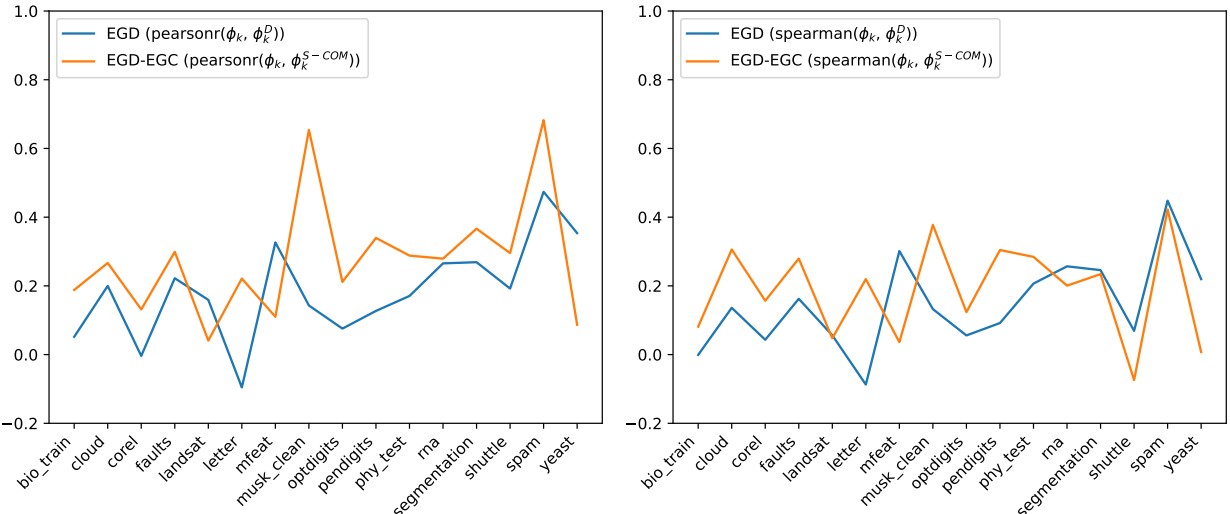

Figure S2: k-means**: Correlations between $(\Phi_K, \Phi_K^{\mathbf{D}})$ and $(\Phi_K, \Phi_K^{\mathbf{S\text{-}COM}})$ on the datasets from (Celebi et al., 2013)**. Correlations are computed from the values of 150 repeats of k-means, using as initialization k-means++-G for $\Phi_K^{\mathrm{D}}$ and Seeding-EGD-EGC for $\Phi_K^{\mathrm{S\text{-}COM}}$.

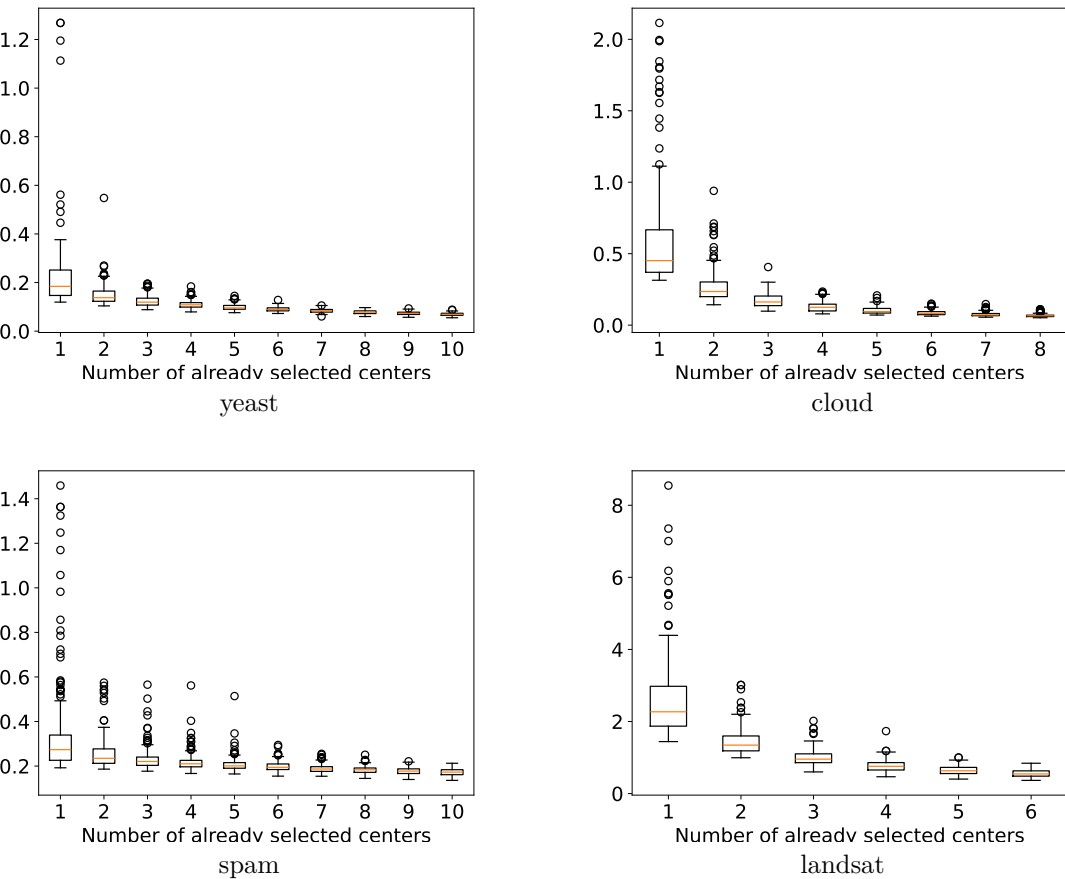

Figure S3: `k-means`**: boxplot of the mean square distance** $\overline{D}^2$ **(each sample to its nearest seed) along the seeding selection process** $- k \in 1..K$**.** Statistics over 150 repeats of `k-means++` on each dataset.

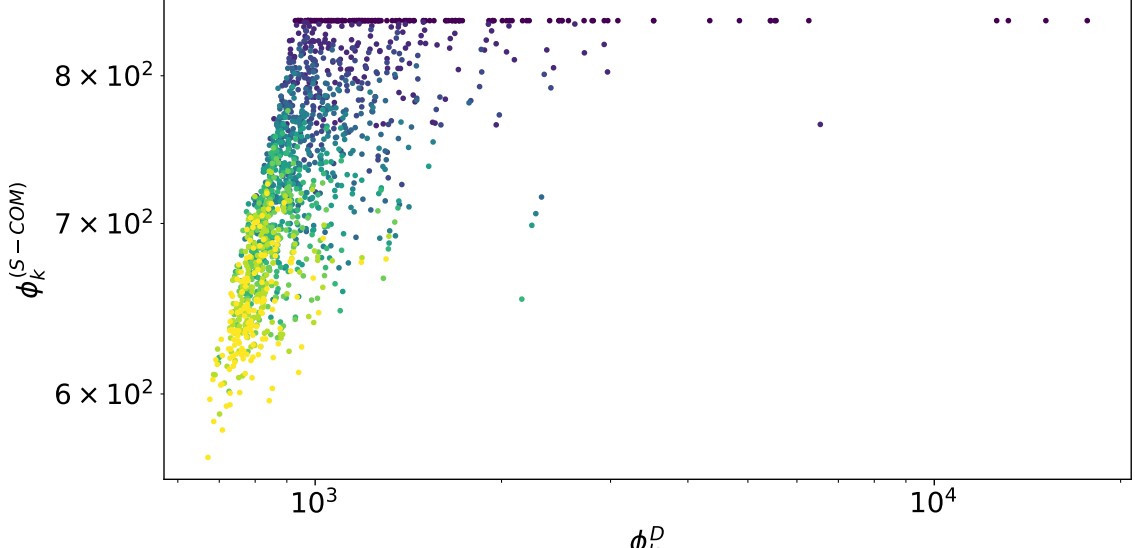

Figure S4: `k-means`: **scatterplot of the $\Phi_K^{\mathbf{D}}$ and $\Phi_K^{\mathbf{S\text{-}COM}}$ values along the seeding selection process for each $k \in 1..K$.** Statistics over 150 repeats on *spam* dataset. The darker the dot, the earlier in the selection process the values were measured.

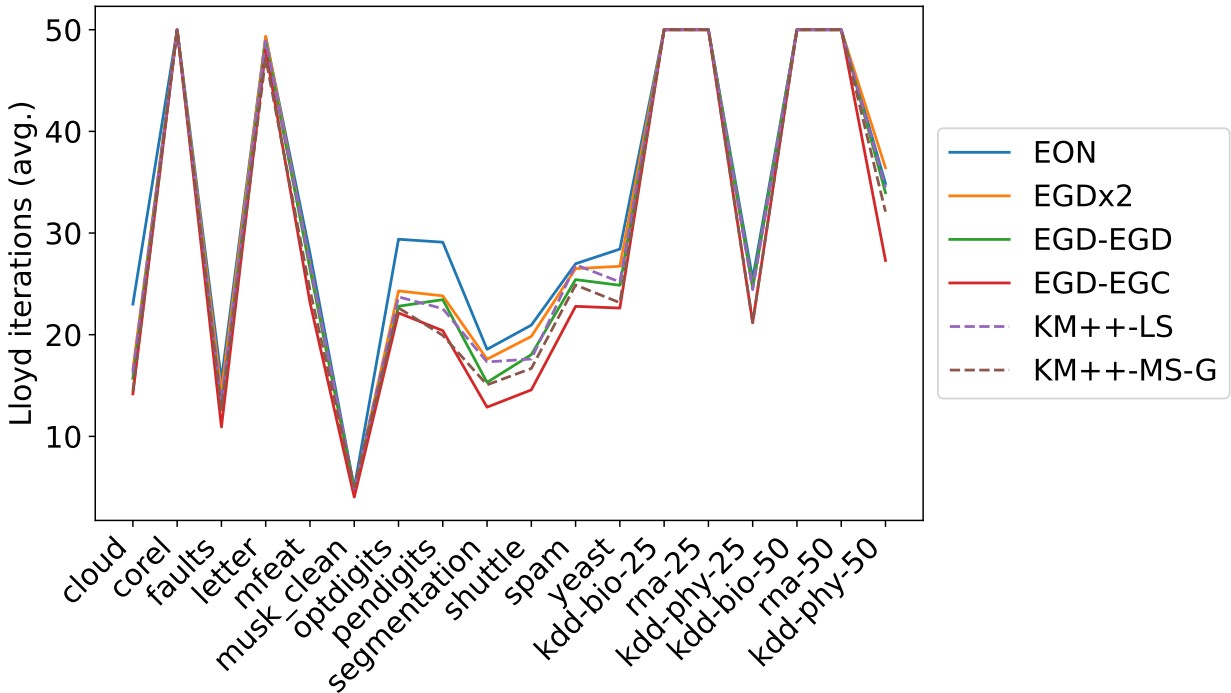

Figure S5: `k-means`: **incidence of the seeding method on the average number of Lloyd iterations.**

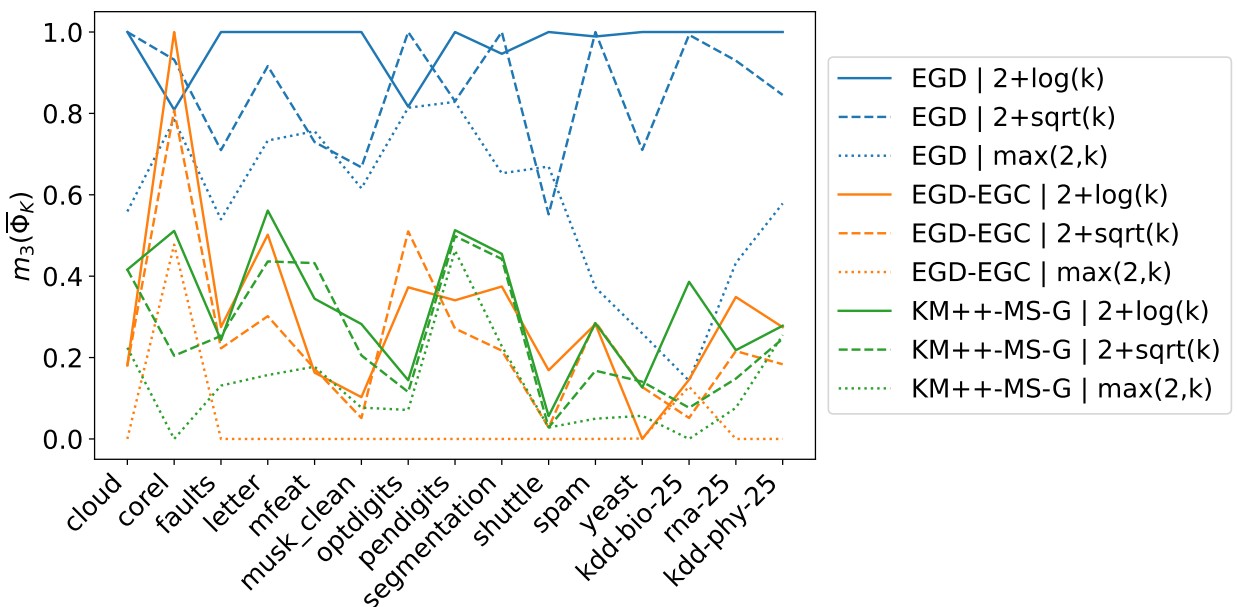

Figure S6: `k-means`: **min-max normalized** $m_3(\overline{\Phi}_K) - $ **Eq.** (4)**, as a function of the seeding method and the candidate pool size.**

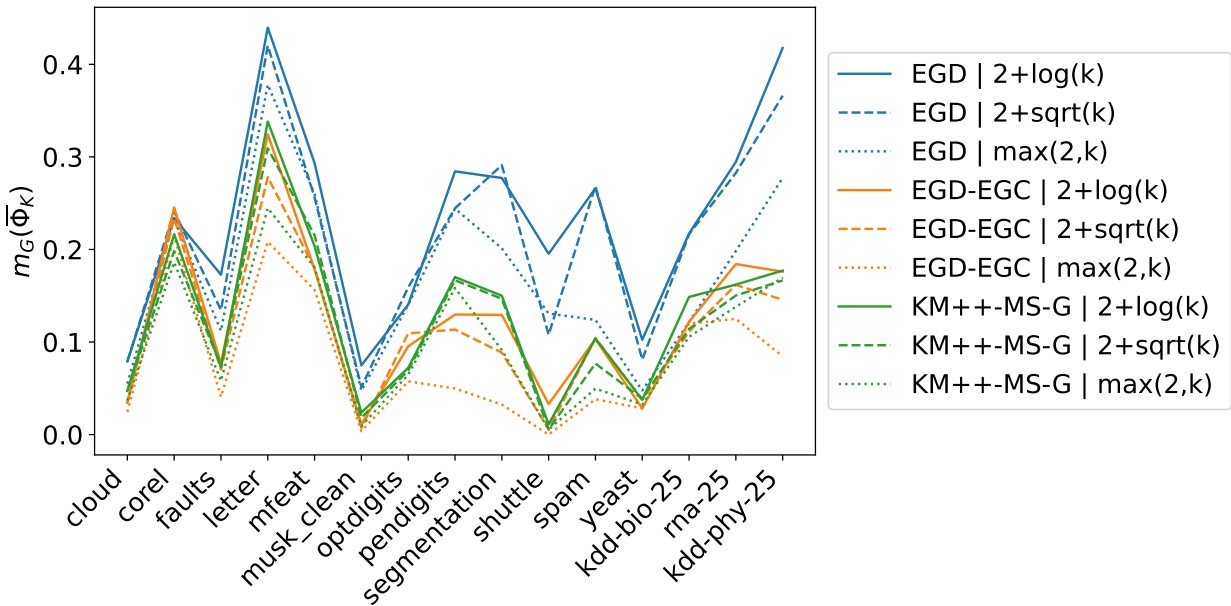

Figure S7: `k-means`: **min-max normalized** $m_G(\overline{\Phi}_K) - $ **Eq.** (5)**, as a function of the seeding method and the candidate pool size.**

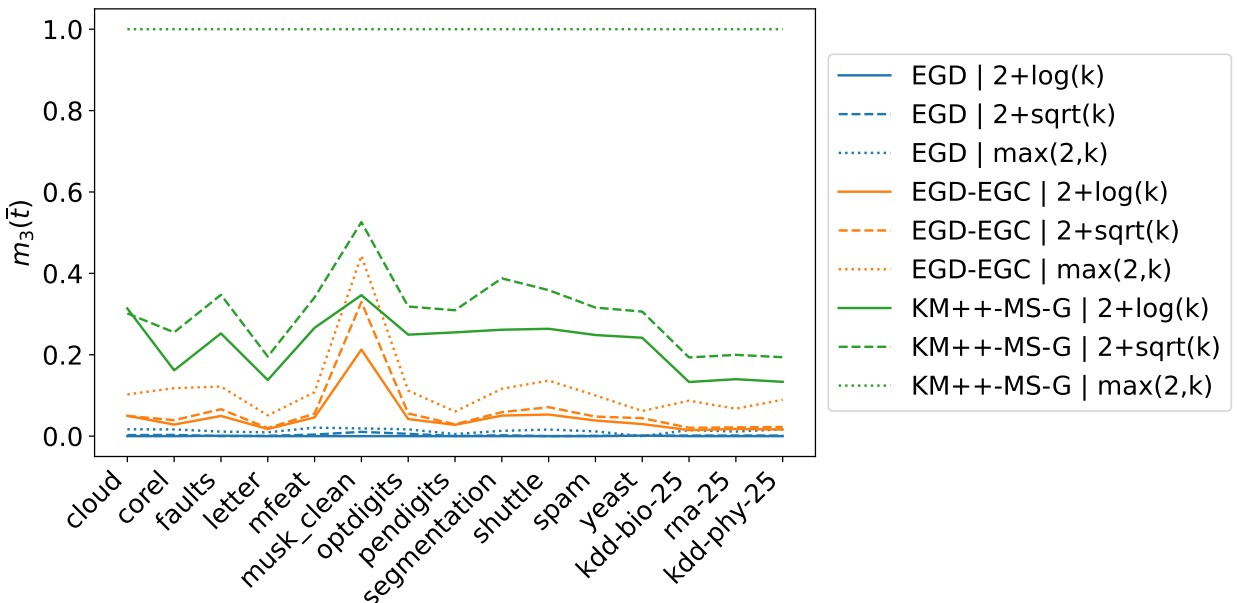

Figure S8: `k-means`: **min-max normalized CPU time** $m_3(\bar{t})$ **as a function of the seeding method and the candidate pool size.**

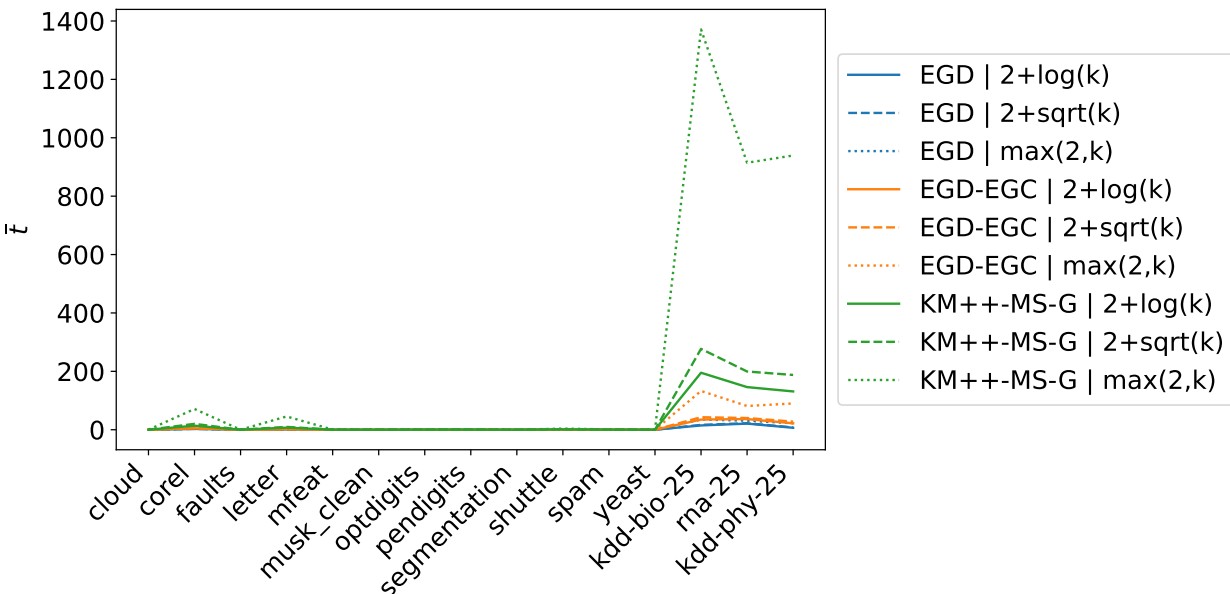

Figure S9: `k-means`: **Raw CPU time (in seconds) as a function of the seeding method and the candidate pool size.**

| | | EON | EGDx2 | EGD-EGD | EGD-EGC | KM++-LS | KM++-MS-G |
|---|---|---|---|---|---|---|---|
| cloud | min | 123.52 | 123.52 | 123.52 | 123.52 | 123.52 | 123.52 |
| | mean | 130.90 ± 10.29 | 125.13 ± 2.06 | 124.93 ± 1.75 | **124.78 ± 1.36** | 126.18 ± 6.22 | 125.25 ± 2.13 |
| corel | min | 10060.20 | 10058.70 | 10057.90 | 10058.60 | 10058.20 | 10057.30 |
| | mean | 10095.31 ± 21.07 | 10095.15 ± 22.22 | 10089.01 ± 22.33 | **10087.06 ± 18.82** | 10095.81 ± 25.61 | 10091.28 ± 22.49 |
| faults | min | 1260.81 | 1260.74 | 1260.75 | 1260.74 | 1260.76 | 1260.75 |
| | mean | 1318.25 ± 48.02 | 1286.21 ± 25.95 | 1283.32 ± 23.15 | **1275.09 ± 15.21** | 1287.80 ± 30.52 | 1275.91 ± 16.48 |
| letter | min | 2723.84 | 2721.15 | 2715.48 | 2718.34 | 2719.92 | 2715.45 |
| | mean | 2756.12 ± 19.12 | 2752.34 ± 17.83 | 2745.91 ± 15.16 | **2743.34 ± 12.88** | 2754.10 ± 17.61 | 2744.54 ± 14.39 |
| mfeat | min | 3128.00 | 3126.49 | 3127.66 | 3128.07 | 3128.64 | 3127.65 |
| | mean | 3171.67 ± 32.60 | 3149.32 ± 18.61 | 3148.54 ± 18.64 | **3142.10 ± 12.47** | 3165.01 ± 28.88 | 3144.34 ± 16.01 |
| musk-clean | min | 36372.70 | 36372.70 | 36372.70 | 36372.70 | 36372.70 | 36372.70 |
| | mean | 37086.86 ± 1319.74 | 36953.47 ± 1288.51 | 36639.47 ± 661.18 | **36448.92 ± 373.40** | 36677.58 ± 698.57 | 36563.25 ± 571.65 |
| optdigits | min | 14559.20 | 14559.20 | 14559.20 | 14559.20 | 14559.20 | 14559.20 |
| | mean | 14708.80 ± 207.68 | 14655.94 ± 116.61 | 14649.75 ± 121.71 | 14625.54 ± 80.60 | 14685.01 ± 158.61 | **14609.40 ± 85.33** |
| pendigits | min | 4930.15 | 4930.15 | 4930.15 | 4930.15 | 4930.15 | 4930.15 |
| | mean | 5073.43 ± 109.36 | 5052.98 ± 81.65 | 5029.72 ± 67.03 | **4990.43 ± 68.30** | 5047.08 ± 92.85 | 5009.22 ± 52.44 |
| segmentation | min | 386.98 | 386.98 | 386.98 | 386.98 | 386.98 | 386.98 |
| | mean | 410.17 ± 16.20 | 399.51 ± 12.64 | 399.29 ± 12.07 | **392.31 ± 8.57** | 401.62 ± 12.63 | 395.82 ± 11.54 |
| shuttle | min | 234.98 | 234.98 | 234.98 | 234.98 | 234.98 | 234.98 |
| | mean | 263.56 ± 35.45 | 243.63 ± 17.91 | 238.48 ± 11.15 | **235.37 ± 3.88** | 238.48 ± 11.15 | **235.37 ± 3.88** |
| spam | min | 526.08 | 525.21 | 524.80 | 524.78 | 525.04 | 524.80 |
| | mean | 566.65 ± 18.80 | 534.99 ± 8.64 | 533.66 ± 8.72 | **531.39 ± 7.07** | 556.24 ± 16.64 | 531.43 ± 7.37 |
| yeast | min | 58.33 | 58.28 | 58.28 | 58.28 | 58.28 | 58.28 |
| | mean | 63.59 ± 5.38 | 58.80 ± 0.45 | 58.74 ± 0.47 | **58.62 ± 0.37** | 62.31 ± 5.04 | 58.74 ± 0.37 |
| kdd-bio-25 | min | 28534.60 | 28546.60 | 28540.90 | 28534.70 | 28548.80 | 28544.50 |
| | mean | 28643.24 ± 132.45 | 28624.17 ± 111.63 | 28592.46 ± 65.25 | **28589.99 ± 46.53** | 28666.10 ± 151.53 | 28602.17 ± 83.51 |
| kdd-phy-25 | min | 137276.00 | 137367.00 | 137301.00 | 136745.00 | 138237.00 | 136783.00 |
| | mean | 144905.69 ± 3654.05 | 140447.92 ± 1438.83 | 139638.87 ± 1239.44 | **138522.96 ± 919.62** | 142497.14 ± 2843.84 | 138536.81 ± 792.31 |
| rna-25 | min | 16690.40 | 16661.20 | 16654.50 | 16652.40 | 16668.50 | 16653.50 |
| | mean | 16941.96 ± 154.55 | 16823.94 ± 105.86 | 16795.23 ± 93.95 | 16770.02 ± 85.54 | 16877.44 ± 129.90 | **16755.85 ± 74.52** |
| kdd-bio-50 | min | 26609.40 | 26616.10 | 26614.60 | 26609.40 | 26606.40 | 26613.20 |
| | mean | 26659.91 ± 32.01 | 26657.06 ± 28.52 | 26656.56 ± 26.37 | **26640.41 ± 15.77** | 26662.84 ± 30.81 | 26642.95 ± 19.02 |
| kdd-phy-50 | min | 116132.00 | 113349.00 | 112627.00 | 112559.00 | 115498.00 | 112487.00 |
| | mean | 119015.89 ± 1958.53 | 115251.90 ± 849.30 | 114688.41 ± 790.79 | 113655.36 ± 536.03 | 117959.49 ± 1662.75 | **113496.50 ± 483.98** |
| rna-50 | min | 12568.40 | 12552.20 | 12551.40 | 12517.40 | 12546.40 | 12530.20 |
| | mean | 12719.97 ± 105.27 | 12633.19 ± 48.48 | 12617.03 ± 41.81 | 12603.64 ± 32.61 | 12677.79 ± 90.82 | **12600.74 ± 37.07** |

Table S1: k-means seeding: $\Phi_K$ values for each seeding method

|  |  | EON | EGDx2 | EGD-EGD | EGD-EGC | KM++-LS | KM++-MS-G |
|---|---|---|---|---|---|---|---|
| cloud | init | 0.001 | 0.004 | 0.009 | 0.012 | 0.012 | 0.041 |
|  | total | 0.012 | 0.012 | 0.015 | 0.018 | 0.019 | 0.047 |
| corel | init | 0.124 | 0.817 | 2.296 | 2.665 | 4.169 | 15.271 |
|  | total | 2.448 | 3.295 | 4.773 | 5.142 | 6.617 | 17.761 |
| faults | init | 0.001 | 0.004 | 0.011 | 0.013 | 0.013 | 0.046 |
|  | total | 0.010 | 0.013 | 0.018 | 0.019 | 0.021 | 0.053 |
| letter | init | 0.035 | 0.194 | 0.741 | 0.792 | 1.576 | 6.008 |
|  | total | 0.886 | 1.042 | 1.581 | 1.654 | 2.466 | 6.869 |
| mfeat | init | 0.002 | 0.009 | 0.026 | 0.028 | 0.037 | 0.126 |
|  | total | 0.031 | 0.037 | 0.055 | 0.053 | 0.067 | 0.153 |
| musk_clean | init | 0.003 | 0.010 | 0.046 | 0.063 | 0.027 | 0.117 |
|  | total | 0.014 | 0.020 | 0.059 | 0.074 | 0.038 | 0.133 |
| optdigits | init | 0.005 | 0.027 | 0.078 | 0.081 | 0.106 | 0.360 |
|  | total | 0.083 | 0.095 | 0.146 | 0.145 | 0.178 | 0.431 |
| pendigits | init | 0.006 | 0.026 | 0.072 | 0.081 | 0.106 | 0.402 |
|  | total | 0.108 | 0.110 | 0.150 | 0.152 | 0.180 | 0.476 |
| segmentation | init | 0.001 | 0.005 | 0.013 | 0.014 | 0.016 | 0.052 |
|  | total | 0.013 | 0.016 | 0.023 | 0.022 | 0.026 | 0.061 |
| shuttle | init | 0.025 | 0.114 | 0.315 | 0.356 | 0.365 | 1.333 |
|  | total | 0.309 | 0.395 | 0.573 | 0.568 | 0.609 | 1.580 |
| spam | init | 0.004 | 0.018 | 0.056 | 0.058 | 0.079 | 0.267 |
|  | total | 0.065 | 0.077 | 0.116 | 0.108 | 0.142 | 0.329 |
| yeast | init | 0.001 | 0.003 | 0.009 | 0.009 | 0.013 | 0.043 |
|  | total | 0.014 | 0.014 | 0.019 | 0.019 | 0.023 | 0.053 |
| kdd-bio-25 | init | 0.994 | 7.690 | 18.520 | 23.204 | 55.951 | 173.514 |
|  | total | 10.831 | 18.493 | 29.365 | 34.213 | 66.843 | 184.628 |
| rna-25 | init | 0.832 | 4.527 | 15.972 | 16.763 | 34.703 | 124.182 |
|  | total | 18.598 | 22.208 | 34.057 | 34.901 | 52.361 | 142.225 |
| kdd-phy-25 | init | 0.683 | 5.907 | 14.136 | 18.350 | 42.176 | 126.584 |
|  | total | 4.344 | 9.646 | 17.892 | 21.533 | 45.816 | 129.795 |
| kdd-bio-50 | init | 2.233 | 18.795 | 50.984 | 60.093 | 225.736 | 787.681 |
|  | total | 23.265 | 40.878 | 73.338 | 82.949 | 247.151 | 809.553 |
| rna-50 | init | 1.738 | 10.976 | 51.131 | 53.497 | 135.014 | 560.411 |
|  | total | 36.062 | 45.791 | 85.948 | 87.920 | 169.170 | 595.723 |
| kdd-phy-50 | init | 1.447 | 14.326 | 36.560 | 45.919 | 165.633 | 567.335 |
|  | total | 11.803 | 25.323 | 47.206 | 54.071 | 175.971 | 576.968 |

Table S2: `k-means` **seeding: average CPU time (in seconds) for each seeding method.** Init (resp. Total) correspond to seeding (resp. seeding + Lloyd iterations).

## S2.2   `k-GMM`

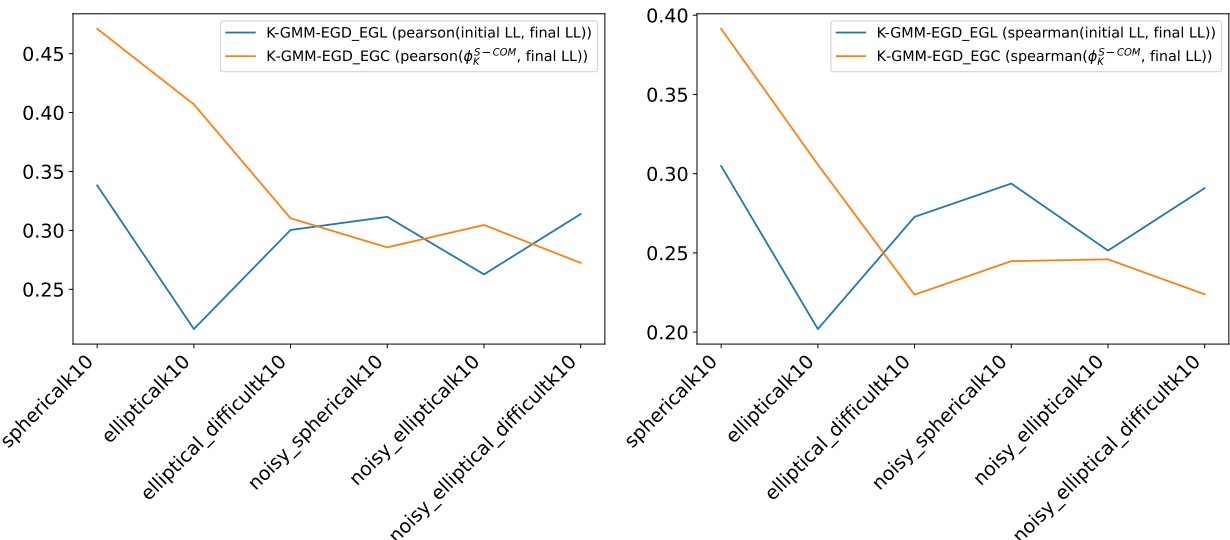

Figure S10:   `k-GMM` **: Correlations (in absolute value) between (initial log-likelihood, final log-likelihood) and ($\Phi_K^{\text{S-COM}}$, final log-likelihood) on the generated datasets from (Blömer & Bujna, 2013).** Correlations are computed from the values of 30 repeats of EM per dataset, using as initialization `K-GMM-seeding-EGD-EGC` for log-likelihood and `K-GMM-seeding-EGD-EGC` for $\Phi_K^{\text{S-COM}}$.

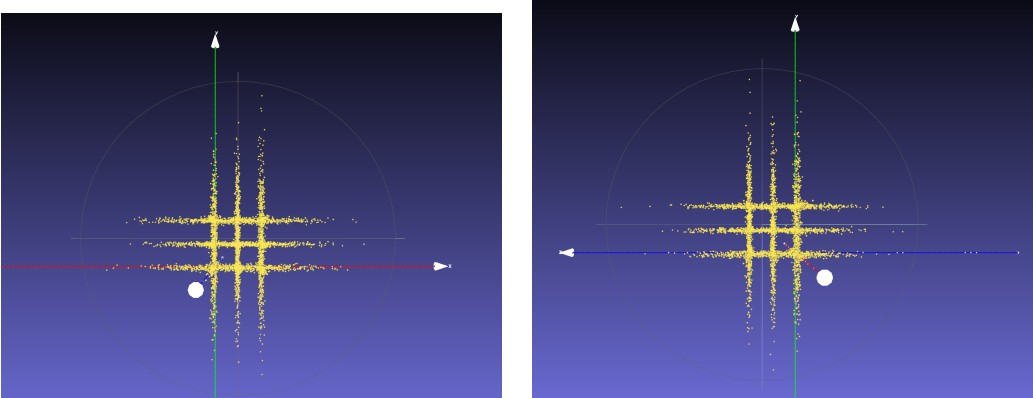

Figure S11: `k-GMM` : **Top and side views of a 3D point cloud generated by a mixture of 27 anisotropic Gaussian distributions.**

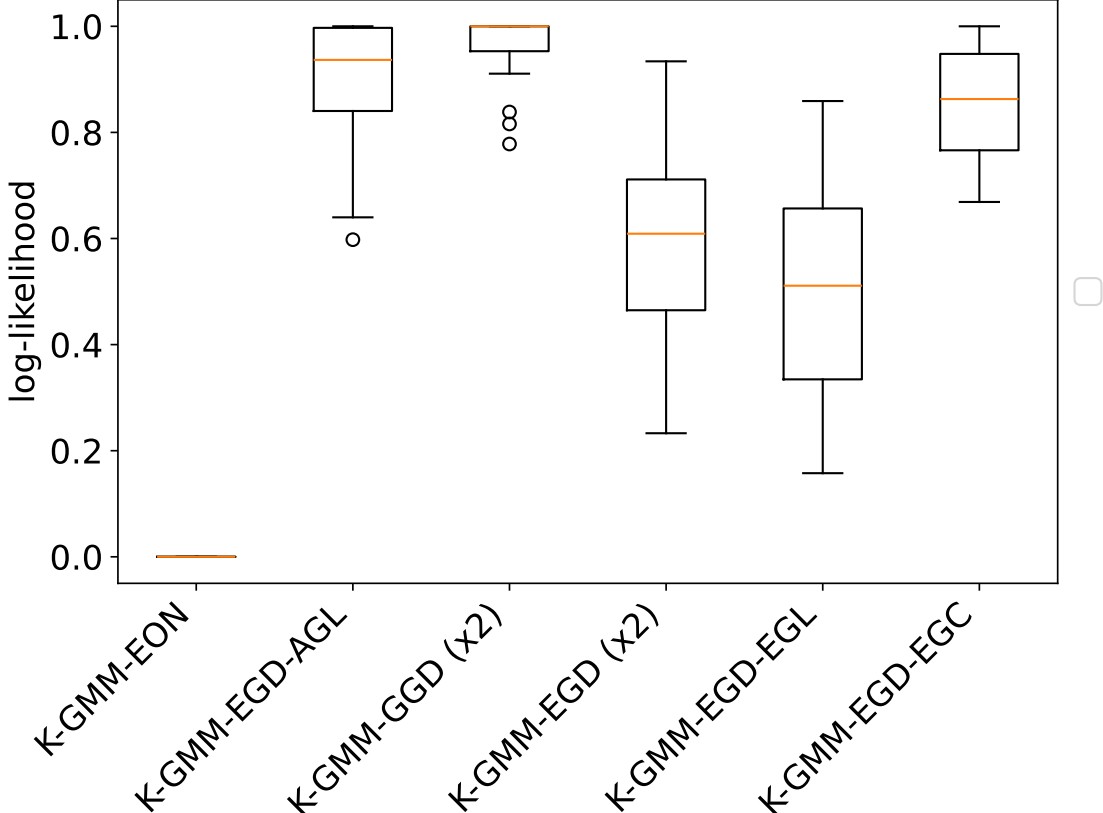

Figure S12: `k-GMM` : **Boxplots of min-max normalized log-likelihoods for the grid dataset.**

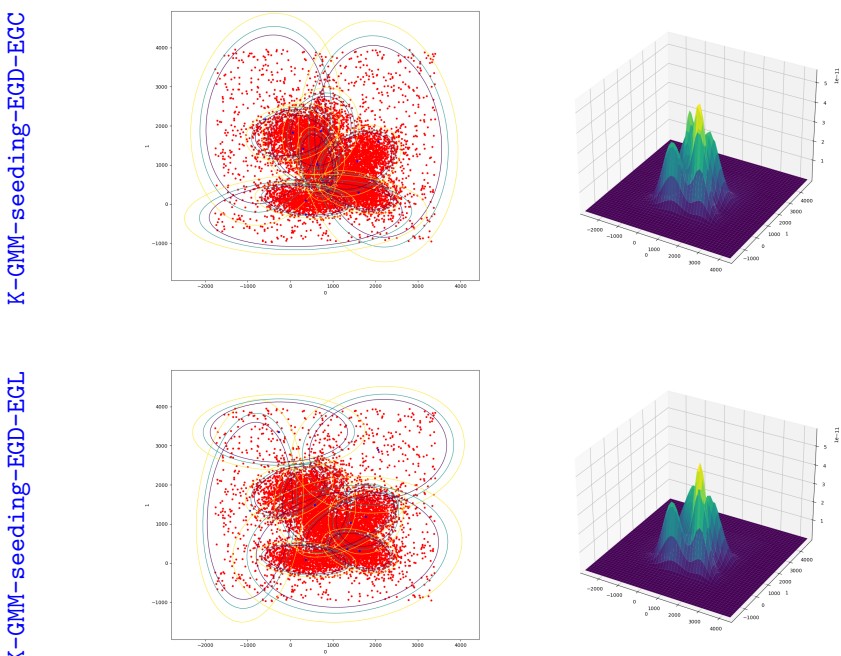

Figure S13: **k-GMM : Fitted models on a *noisy-elliptical* dataset with separation** $s = 0.5$. Data is generated using a GMM with 10 components. Each component is represented with a blue dot for the mean, and contour lines to represent the confidence regions at $[0.85, 0.90, 0.95]$ (darker lines corresponding to higher confidence). Likelihood regulated methods (bottom) are highly sensitive to noise and consistently outperformed by SSE regulated methods (top).

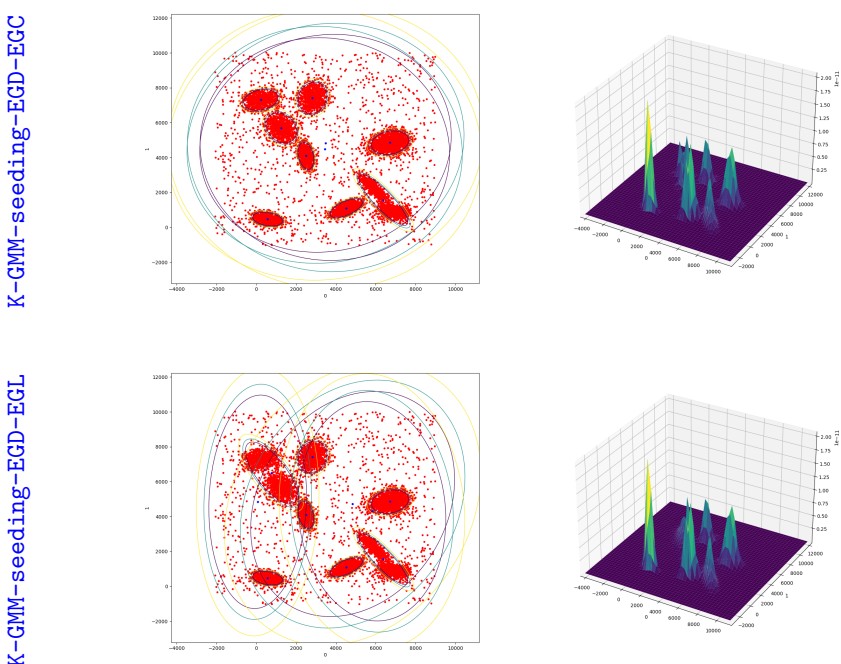

Figure S14: **k-GMM : Fitted models on a *noisy-elliptical* dataset with separation $s = 2$.** Data is generated using a GMM with 10 components. Conventions identical to Fig. S13.

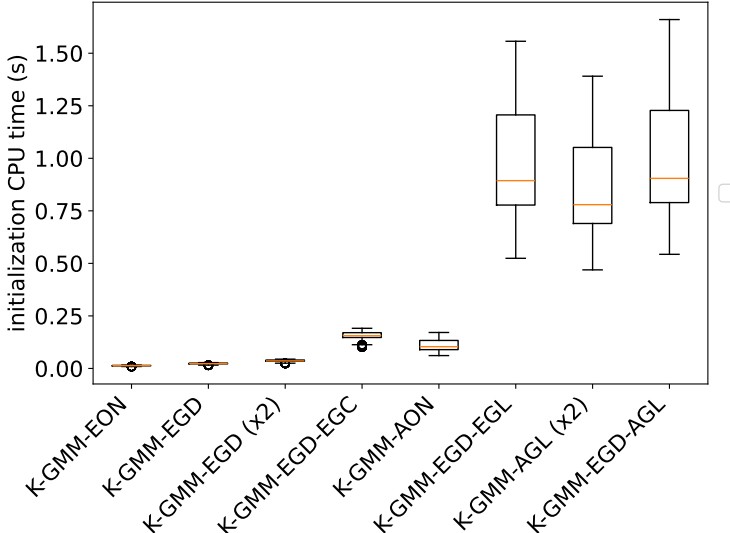

Figure S15: `k-GMM` : **average CPU time in seconds over all datasets for initialization only, when using each seeding method.**

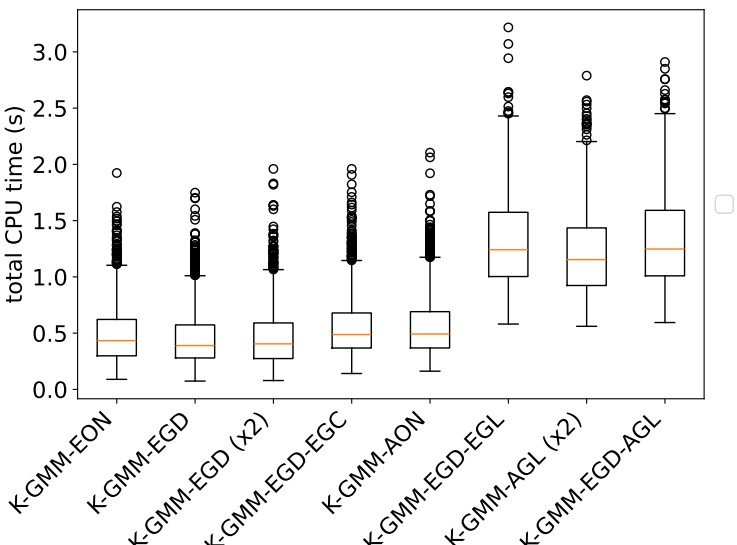

Figure S16: `k-GMM` : **average CPU time in seconds over all datasets for a full EM run, when using each seeding method.**

| | EON | EGD | EGD (x2) | EGD-EGC | EGD-EGL | AGL (x2) | EGD-AGL | AON | GGD (x2) |
|---|---|---|---|---|---|---|---|---|---|
| sphericalk10 | 0.51 ± 0.24 | 0.78 ± 0.18 | 0.83 ± 0.16 | 0.91 ± 0.15 | 0.92 ± 0.13 | 0.92 ± 0.11 | **0.95** ± 0.10 | 0.33 ± 0.23 | 0.77 ± 0.22 |
| ellipticalk10 | 0.47 ± 0.22 | 0.73 ± 0.18 | 0.81 ± 0.17 | **0.93** ± 0.14 | 0.85 ± 0.20 | 0.89 ± 0.14 | **0.93** ± 0.14 | 0.25 ± 0.20 | 0.79 ± 0.16 |
| elliptical-difficultk10 | 0.50 ± 0.23 | 0.74 ± 0.20 | 0.78 ± 0.20 | 0.82 ± 0.24 | 0.88 ± 0.18 | 0.87 ± 0.15 | **0.91** ± 0.14 | 0.19 ± 0.16 | 0.78 ± 0.26 |
| noisy-sphericalk10 | 0.32 ± 0.21 | 0.73 ± 0.20 | 0.79 ± 0.22 | **0.83** ± 0.23 | 0.48 ± 0.29 | 0.65 ± 0.24 | 0.66 ± 0.24 | 0.39 ± 0.20 | 0.78 ± 0.24 |
| noisy-ellipticalk10 | 0.31 ± 0.18 | 0.82 ± 0.12 | 0.91 ± 0.12 | **0.94** ± 0.10 | 0.19 ± 0.21 | 0.45 ± 0.21 | 0.50 ± 0.20 | 0.46 ± 0.26 | 0.87 ± 0.13 |
| noisy-elliptical-difficultk10 | 0.27 ± 0.22 | 0.77 ± 0.16 | 0.86 ± 0.15 | 0.83 ± 0.20 | 0.24 ± 0.24 | 0.51 ± 0.20 | 0.51 ± 0.22 | 0.38 ± 0.25 | **0.89** ± 0.19 |

Table S3: k−GMM : mean and standard deviation of min-max normalized log-likelihood for each seeding method

