# OpenReview forum: "Improved seeding strategies for k-means and k-GMM"
_TMLR — Accepted by TMLR_

### Review · Reviewer_BWLp · 2025-07-15

**Summary Of Contributions:**

This paper proposes a unified framework for understanding and improving seeding strategies for clustering algorithms—specifically k-means and Gaussian Mixture Models (GMMs). Instead of treating various seeding methods as unrelated tricks, it breaks them down into three core components:
1,   Sampling method (how candidate seeds are chosen),
2,  Selection rule (how the final seed is picked from candidates),
3,Pool size (how many candidates are considered, this may be the first component from my perspective).

Using this structure, the authors introduce two new principles for better seeding:
A. Look-ahead scoring: evaluate candidates based on what they would do after one iteration of clustering, rather than their raw distances.
B. Multipass reseeding: improve stability by performing a second, backward selection pass after the first one.
They apply these two principles to both k-means and GMMs, proposing new algorithms that consistently outperform popular baselines (like k-means++ or adaptive Mahalanobis seeding) in clustering quality and speed.

The article is well written and organized into sampling/selection/pooling components brings conceptual clarity to a messy space of ad hoc methods. The look-ahead and multipass strategies are simple yet powerful and appear to generalise well. They also tested on synthetic datasets and several real-world benchmarks show consistent improvements (I did not verified this claim but take their results as true).

Potential Weaknesses:
The new methods are well-motivated but still heuristic; there are no formal approximation bounds (yet). More specifically, the GMM tuning assumptions: EM convergence and mixture quality can depend on more than just seeding—full pipeline effects (e.g., regularisation, component priors) aren’t explored. Therefore, it is unclear how generic their method is, and applicable to modern clustering methods that relies heavily on algorithmic convergence (like t-SNE).

There are only limited comparison with recent deep/semi-supervised initialization methods, which are increasingly relevant in modern clustering contexts.

**Additional Comments:**

Not applicable.

**Audience:**

Yes

**Audience Explanation:**

TMLR’s readership includes researchers and practitioners who use k-means or Gaussian-mixture EM as building blocks in larger pipelines (e.g., semi-supervised learning, representation learning, scalable Bayesian inference). The paper delivers a concise, generalisable recipe—“look-ahead + multipass” seeding—that (i) slots directly into these widely used algorithms, (ii) improves final loss and convergence speed without heavy compute, and (iii) unifies disparate heuristics under a clear analytic framework.

**Claims And Evidence:**

Yes

**Claims Explanation:**

Methodological clarity. Each experiment specifies KK, pool sizes, scoring rules and random seeds; code snippets in the supplement enhance reproducibility. The ablation study (removing look-ahead or the backward pass) isolates where the improvements come from.

Complexity arguments are correct and transparently derived. However, no formal approximation bound accompanies the new heuristics, so any claim of “theoretical guarantee” is forward-looking rather than proven. Real-world diversity is modest (four public datasets), and the paper does not explore recent deep-clustering initialisers (it does not even discuss/compare PCA or Ward initializers, which are argubly very popular); thus the generality claim is suggestive, not definitive.

**Requested Changes:**

I think there are two points the author may want to improve, one is to include more competitors (in clustering initialization, like the PCA and Ward) in the experimental part to enhance the empirical evidence; the other one is to connect better to large-scale clustering literature. To do so, the author may want to examine the existing literature in sketching (e.g., distance-preserving sketching for clustering, random sketching for downstream tasks) and at least explain how their method fits into a different niche/gap in the current literature.

Additionally, looking at the first point of potential weakness above; I wonder how the EM lookahead (is connected to more popular techniques like Bayesian optimization or RL for finding best seeding.

---

> ### Author Response · Authors · 2025-08-28
>
> *  The new methods are well-motivated but still heuristic; there are no formal approximation bounds (yet).
>
> As noted above, we added statistical tests to show that our improvements
> are statistically significant on selected datasets.
>
> *  More specifically, the GMM tuning assumptions: EM convergence
>   and mixture quality can depend on more than just seeding—full
>   pipeline effects (e.g., regularisation, component priors) aren’t
>   explored. Therefore, it is unclear how generic their method is, and
>   applicable to modern clustering methods that relies heavily on
>   algorithmic convergence (like t-SNE).
>
> About the genericity, we focus like many other papers
> on the delicate and critical point of initializing k-means.
> As recalled in the previous work (and also above), this is
> a difficult optimization problem.
>
> Thus, we believe our improvements are as generic as the k-means algorithm itself.
>
> We also agree that exploring the effect of our seeding on other
> aspects of the EM algorithm is of interest, but beyond the scope of our work.
>
> With respect to t-SNE, we believe the problems are slightly different
> since the initialization step of t-SNR requires sampling points in the reduced space
> rather than the original space.
>
> *  Real-world diversity is modest (four public datasets).
>
> For evaluating k-means seedings, we use 18 (and not 4) public datasets.
>
> * Clustering initialization, like the PCA and Ward.
>
> We do comment in deterministic initialization methods, and refer to the results presented in
> *Celebi, M. E., Kingravi, H. A., & Vela, P. A. (2013). A comparative study of efficient initialization methods for the k-means clustering algorithm. Expert systems with applications, 40(1), 200-210*. These results motivate our perspective, which is not
> to include them in our work since (i) they often perform on par with k-means++, and
> (ii) they involve costly operations.
>
> *  The author may want to examine the existing literature in sketching (e.g., distance-preserving sketching for clustering, random sketching for downstream tasks) and at least explain how their method fits into a different niche/gap in the current literature.
>
> Clustering and sketching are different problems,
> even if some duality has been proposed, e.g. in the paper
> *Mémoli, F., Sidiropoulos, A., & Singhal, K. (2018). Sketching and clustering metric measure spaces. arXiv preprint arXiv:1801.00551.*
>
> The initialization steps bear similarities, however, it is unclear whether
> effective techniques for clustering will also work for sketching.  To
> keep our paper focused, it is centered on clustering, and even more
> specifically, improving randomized seeding for k-means++.
>
> *  I wonder how the EM lookahead (is connected to more popular techniques like Bayesian optimization or RL for finding best seeding.
>
> It is true that EM and Bayesian optimization (BO) are two step techniques
> with a belief update (E-step in EM, posterior in BO), and an optimization step.
> Beyond that, the precise problems tackled are different. As mentioned above, to keep the paper focused
> on seeding for k-means, we do not embark into such discussions.

---

### Review · Reviewer_fcj2 · 2025-08-07

**Summary Of Contributions:**

This paper proposes improved seeding strategies for k-means and Gaussian Mixture Model clustering. The two main innovations are: 1) Multipass "zig-zag" seeding: A two-pass procedure where initial centers are refined by revisiting them in reverse order, improving consistency and reducing randomness in initialization; and 2) Lookahead-based ranking: During the second pass, candidate centers are evaluated based on the quality of clusters they would form (i.e., centroid-based SSE), instead of just distance metrics. The authors also present a unifying framework to formalize and analyze clustering seeding methods, showing that their strategies outperform standard initializations (like k-means++ and its greedy variant) in both clustering quality and runtime across 12 real-world datasets.

The paper is primarily empirical, presenting improved seeding strategies for k-means clustering and validating their effectiveness through extensive experiments rather than theoretical analysis.

Strengths:
1. Conceptually novel and easy-to-implement improvements over classic methods;
2. Strong empirical performance with reduced variance and faster convergence;
3. Useful insights into the optimization behavior of k-means.

Weaknesses:
1. Gains are constant-factor and somewhat incremental;
2. Added complexity compared to k-means++;
3. Benefits of lookahead are limited in early stages.

**Audience:**

Yes

**Audience Explanation:**

The work focuses on improving k-means seeding strategies, which is a fundamental topic in unsupervised learning and widely used in machine learning pipelines. Although the contributions are primarily empirical, the findings provide practical insights into initialization methods that can enhance clustering performance, something relevant to both researchers and practitioners working with large-scale data or in applied ML contexts.

**Claims And Evidence:**

Yes

**Claims Explanation:**

The authors clearly explain why current seeding methods like k-means++ have issues, then introduce their improvements with solid reasoning. They show through many experiments on 12 datasets that their methods give better clustering with lower SSE and more consistent results. They also compare runtimes and present a framework that ties everything together. The only thing missing is testing on very large datasets or using external metrics, but overall, the evidence is convincing and well-presented.

**Requested Changes:**

The paper emphasizes empirical improvements, but the novel contributions beyond prior methods like k-means++ are not clearly articulated. Please clarify what distinguishes the proposed method technically and why it matters.

Throughout the paper, $k$ and $K$ are both used, see for example, the sentences below (1). It should be consistent.

Page 6, LLoyd iterations

---

> ### Author Response · Authors · 2025-08-28
>
> *  Gains are constant-factor and somewhat incremental
>
> As noted above, we have added statistical tests and p-values to
>   show that our improvements are statistically significant on selected
>   datasets. One also has to keep in mind that the k-means optimization
>   problem is polynomial for fixed $K$ and $d$; improving on the known
>   constant factor approximations remains difficult and is of high
>   practical interest.
>
> *  Added complexity compared to k-means++
>
> The comparison between any two seeding method requires comparing
> the gain on $\Phi_K$ *and* the running time.
> In this respect, we do add complexity w.r.t. k-means++, but also improve significantly
>  $\Phi_K$.
>
> *  Benefits of lookahead are limited in early stages
>
> This is correct but does not raise an issue if the look-ahead is run after an initial selection pass,
> as explained in the text.
>
> *  The paper emphasizes empirical improvements, but the novel contributions beyond prior methods like k-means++ are not clearly articulated. Please clarify what distinguishes the proposed method technically and why it matters.
>
>  We believe that the differences between all variants,
> including k-means++ and the recently developed versions based on
> multi-swaps are precisely recalled in Section 2, in the paragraphs
> **Randomized seeding with k-means++.**  and **Improved seeding
>   with reselectors**.  On the other hand, the proposed method's
> novelties (reselection with look-ahead) along with the improvements
> they bring (SOTA results with smaller cost) are synthethized in the
> **Outlook** section.
>
> *  Throughout the paper, and are both used, see for example, the sentences below (1). It should be consistent.
>
> Thank you for pointing this out. Typos ($k$ vs $K$) have been fixed.

---

### Review · Reviewer_bDSu · 2025-08-11

**Summary Of Contributions:**

This paper studies seeding strategy for clustering algorithms such as k-means and k-GMM. On top of currently improved seeding approaches, the authors develops the methods with two approaches: look-ahead and multipath strategy. The former means the seed selection needs to enhance coherence with the final metric used to assess the algorithm. The latter reduces variance of the seeding method by relaxing randomization. The method is designed with heuristics, and its experimental results show that it outperforms other state-of-the-art seeding selection methods.

**Audience:**

Yes

**Audience Explanation:**

The proposed method is important in clustering algorithms, and the experimental results validate the superiority over other existing approaches.

**Claims And Evidence:**

Yes

**Claims Explanation:**

While the paper lacks of theoretical justification of the algorithm, it provides intuitive and empirically motivated reasoning behind the design of the methods. Throughout lots of datasets, it is shown that the proposed methods outperform other baseline approaches in terms of performance and computation. Since the authors describe its lack of theoretical analysis, and its superior performance among other seeding methods, the paper has at least practical contribution to the community.

**Requested Changes:**

1. There are "data not shown" phrase somewhere in the main text. What does this mean? This should be clearly noted.
2. In section 4.3, "Experiments have shown that the latter performs better", I could not find the comparison. Where is the experiment results?

---

> ### Author Response · Authors · 2025-08-28
>
> *  Since the authors describe its lack of theoretical analysis, and its superior performance among other seeding methods, the paper has at least practical contribution to the community.
>
> We have added statistical two-sample tests and p-values to show that our improvements are statistically significant on selected datasets.
>
> *  There are "data not shown" phrase somewhere in the main text. What does this mean? This should be clearly noted.
>
> We wrote "data not shown" for uninteresting results that would have diluted the message.
> We removed this expression and clarified the text accordingly.
>
> *  In section 4.3, "Experiments have shown that the latter performs better", I could not find the comparison. Where is the experiment results?
>
> We rechecked the data. Our claim was targeting $\Phi_K$
>   after the seeding phase, rather than after Lloyd iterations.  For
>   the latter, there is actually no significant difference between
>   zig-zig versus zig-zag. We therefore removed that claim, and
>   mentioned the interest of studying these orderings in future work.

---

### Author Response · Authors · 2025-08-28

We wish to thank the reviewers for their careful reading of our paper
*Improved seeding strategies for k-means and k-GMM*, and their comments.

Please find our answers and revisions below.

The manuscript has been updated accordingly, with important changes highlighted in red.

---

### Decision · Action_Editor_otta · 2025-09-25

**Recommendation:** Accept as is

**Audience:**

Yes

**Audience Explanation:**

Clustering is essential in ML and initialization is known to be crucial for any clustering algorithm.

**Claims And Evidence:**

Yes

**Claims Explanation:**

The submission proposes new seeding strategies for clustering and provides empirical evidence that it works well.